# ATP functions as a pathogen-associated molecular pattern to activate the E3 ubiquitin ligase RNF213

Juraj Ahel [1], Arda Balci[2], Victoria Faas[1,3], Daniel B. Grabarczyk [1], Roosa Harmo[2], Daniel R. Squair[4], Jiazhen Zhang[4], Elisabeth Roitinger [5], Frederic Lamoliatte[4], Sunil Mathur[4], Luiza Deszcz[1], Lillie E. Bell [1,3], Anita Lehner[6], Thomas L. Williams[1], Hanna Sowar[2], Anton Meinhart[1], Nicola T. Wood [4], Tim Clausen [1,7] ✉, Satpal Virdee [4] ✉ & Adam J. Fletcher [2] ✉

The giant E3 ubiquitin ligase RNF213 is a conserved component of mammalian cell-autonomous immunity, limiting the replication of bacteria, viruses and parasites. To understand how RNF213 reacts to these unrelated pathogens, we employ chemical and structural biology to find that ATP binding to its <u>A</u>TPases <u>A</u>ssociated with diverse cellular <u>A</u>ctivities (AAA) core activates its E3 function. We develop methodology for proteome-wide E3 activity profiling inside living cells, revealing that RNF213 undergoes a reversible switch in E3 activity in response to cellular ATP abundance. Interferon stimulation of macrophages raises intracellular ATP levels and primes RNF213 E3 activity, while glycolysis inhibition depletes ATP and downregulates E3 activity. These data imply that ATP bears hallmarks of a danger/pathogen associated molecular pattern, coordinating cell-autonomous defence. Furthermore, quantitative labelling of RNF213 with E3-activity probes enabled us to identify the catalytic cysteine required for substrate ubiquitination and obtain a cryo-EM structure of the RNF213-E2-ubiquitin conjugation enzyme transfer intermediate, illuminating an unannotated E2 docking site. Together, our data demonstrate that RNF213 represents a new class of ATP-dependent E3 enzyme, employing distinct catalytic and regulatory mechanisms adapted to its specialised role in the broad defence against intracellular pathogens.

The innate immune system represents the earliest host barrier to invading pathogens. A series of genetically encoded sensor proteins called pattern recognition receptors (PRRs) detect specific molecular signatures – pathogen-associated molecular patterns (PAMPs) – that distinguish pathogen from the host; exemplar PAMPs include viral nucleic acids[1] and bacterial peptidoglycans[2]. PRRs can also detect host-derived molecules aberrantly localized during infection, referred to as damage-associated molecular patterns (DAMPs); exemplar DAMPs include host cell DNA[3], ATP[4] and oxidized lipids[5]. PAMP or DAMP detection leads to the production of interferons (IFNs), which induce the expression of hundreds of antimicrobial genes that network to restrict pathogen spread[6] and prime adaptive immunity[7].

[1]Research Institute of Molecular Pathology (IMP), Vienna BioCenter, Vienna, Austria. [2]MRC University of Glasgow Centre for Virus Research, University of Glasgow, Glasgow, United Kingdom. [3]Vienna Biocenter PhD Program, a Doctoral School of the University of Vienna and the Medical University of Vienna, Vienna, Austria. [4]MRC Protein Phosphorylation and Ubiquitylation Unit, School of Life Sciences, University of Dundee, Dundee, United Kingdom. [5]Institute of Molecular Biotechnology (IMBA), Vienna BioCenter, Vienna, Austria. [6]Vienna BioCenter Core Facilities, Vienna BioCenter, Vienna, Austria. [7]Medical University of Vienna, Vienna, Austria. ✉e-mail: tim.clausen@imp.ac.at; s.s.virdee@dundee.ac.uk; adam.fletcher@glasgow.ac.uk

Understanding mechanisms underlying PRR activation and IFN-mediated pathogen restriction are fundamental research objectives, because defects in PRR machinery promote viral pathogenesis, hyperinflammation and interferonopathies[8–10]. Furthermore, understanding PRR activation is contributing to next-generation antiviral and vaccine design[11]. Importantly, several PRRs collaborate with – or are themselves – E3 ubiquitin (Ub) ligases, which permits the coupling of pathogen detection with downstream signaling[12].

RNF213 is the largest E3 Ub ligase (E3) in the human proteome, combining E3 and AAA (ATPases Associated with diverse Activities) modules into a single ~600 kDa polypeptide. RNF213 represented the first example of an E3 targeting a non-proteinaceous substrate[13], and was soon joined by HOIL-1 (sugars)[14] and TUL1 (lipids)[15]. RNF213 is implicated in several pathways, including angiogenesis, fatty acid metabolism and non-mitochondrial oxygen consumption[16–20]. RNF213 has also emerged as an important component of the innate immune system. During *Salmonella* Typhimurium infection, RNF213 detects and ubiquitinates lipopolysaccharide (LPS), a lipid-polysaccharide anchored in the bacterial outer membrane; this promotes pathogen degradation via xenophagy[13]. Further studies have shown that RNF213 has unusually broad antimicrobial activity, limiting the replication of gram-positive and gram-negative bacteria, eukaryotic parasites, and both RNA and DNA viruses[21–26]. In several cases, RNF213 forms a Ub-positive coat around the pathogen-containing vacuole[13,23,24]. This suggests that, in addition to bacterial LPS, RNF213 ubiquitinates unidentified proteins or lipids, and these modifications restrict pathogen growth[21,27]. Such breadth of microbial specificity is unusual for an antimicrobial E3, raising the question of how RNF213 E3 activity is regulated by such diverse pathogen infections.

The cryo-EM structure of *M. musculus* RNF213[28] revealed that RNF213 comprises three sub-modules: an N-terminal stalk, a dynein-like AAA core consisting of six ATPase units (only the two units AAA3 and AAA4 can bind and hydrolyze ATP), and a composite E3 module, including a RING domain (Fig. 1a). The combination of an AAA module and E3 is unprecedented in Ub biology, and the question of whether binding or hydrolysis of nucleoside phosphates regulates RNF213 E3 activity remains unexplored. Furthermore, as LPS and proteins are chemically distinct, its Ub transfer mechanism may have diverged from the canonical RING, HECT, and RBR E3 classes. RING E3s act like scaffolds that position the Ub-loaded E2 conjugating enzyme (E2-Ub) in such a way as to allow the E2 active site to transfer Ub directly to substrate (a process termed aminolysis)[29]. In contrast, for HECT and RBR E3 ligases the E2-Ub delivers Ub onto the E3 active site cysteine (a process termed transthiolation or transthioesterification) before Ub transfer to substrate[30,31]. Regardless of the overarching mechanism, canonical E3s conjugate Ub to the ε-amino group of lysine residues forming an isopeptide bond.

MYCBP2 has emerged as an atypical transthiolating E3 that employs an RCR (RING-Cys-Relay) mechanism to transfer Ub to hydroxy groups of threonine and serine residues in model substrates[32]. This demonstrates that E3s can modify unconventional substrates, and that other transthiolating E3s exhibiting non-lysine specificity are likely to exist[33]. RNF213 may represent such an example. For example, the RNF213 RING is dispensable for autoubiquitination or LPS ubiquitination by full-length protein[13,28]. Furthermore, RNF213 is active with the E2 UBE2L3[28], which cannot perform aminolysis[31,34]. Together, these two observations suggest that RNF213 contains an uncharacterized transthiolating E3 module. Mutation of a conserved histidine in the poorly understood RNF213 RZ (RNF213/ZNFX1) Zn-finger, a flexible domain located within the larger E3 module, abolishes ubiquitination activity[13], suggesting the RZ domain is a component of a cryptic Ub transfer mechanism. Here, we employ chemical activity-based probes and develop innovative assays to dissect RNF213 E3 activation in vitro and in living cells. We find that RNF213 constitutes a new class of transthiolating E3 enzyme activated by cellular nucleotides, and we show

that cellular RNF213 activity can be controlled by adjusting nucleotide abundance. A cryo-EM structure of a covalently-trapped E2-E3 complex confirms the novelty of this class and provides insights into its unusual Ub transfer mechanism.

## Results

### ATP binding stimulates RNF213 E3 Ub ligase activity

Mutation of the ATPase active sites in RNF213 has been reported to affect its ability to oligomerise[35], to associate with intracellular lipid droplets[27], and to target bacterial pathogens[13]. To investigate whether ATPase activity is coupled to E3 activity, we disrupted adenosine-5′-triphosphate (ATP) binding or hydrolysis at the two functional ATPase subunits AAA3 and AAA4 of murine RNF213[28]. K2387A (WA3) or K2736A (WA4) mutations in the Walker A motifs disrupt ATP binding, whereas isosteric E2449Q (WB3) or E2806Q (WB4) mutations in the Walker B motifs disrupt ATP hydrolysis (Fig. 1a)[36]. When tested in autoubiquitination assays, RNF213 WA3 and WA4 variants had substantially reduced activity in the presence of ATP, unlike WB3 and WB4, which were unaffected (Fig. 1b, c). This outcome was mirrored when using the previously reported LPS model substrate, Lipid A[13] (Supplementary Figs. 1a–c). Suggestive of O-linked ubiquitination, the Lipid A ubiquitination product was unstable at high pH ( > 10) (Supplementary Fig. 1c). Together, these data indicate that ATP binding rather than hydrolysis is required to activate the RNF213 E3, regardless of whether the substrate is proteinaceous or non-proteinaceous.

To examine nucleotide regulation of RNF213 without interference from the ATP-dependence of the E1 enzyme, we purified E2 enzymatically loaded with Ub, then followed RNF213-catalyzed Ub discharge from E2-Ub in the presence of various nucleotides. For wild type (WT) RNF213, E3 activity was enhanced by ATP, but not by ADP or AMP (Fig. 1d, e). Consistent with ATP binding stimulating RNF213 activity, the non-hydrolyzable ATP analogues adenosine 5′-O-(3-thiotriphosphate) (ATPγS) and adenylyl-imidodiphosphate (AMP-PNP) also stimulated E2-Ub discharge (Fig. 1d–f). The ATP analogue adenylyl-methylenediphosphonate (AMP-PCP) was an exception, which may have failed due to its reduced affinity for ATP-binding enzymes[37–39] (Fig. 1d, e). Alternative nucleoside triphosphates, such as GTP, CTP and UTP, also failed to stimulate RNF213 activity (Supplementary Figs. 1d, e). Thus, ATP binding to AAA3 and AAA4 subunits is necessary and sufficient to promote the E3 ligase activity of RNF213.

We next sought to determine which step of the RNF213 E3 catalytic cycle is regulated by ATP binding. As prior data (RING domain independence and activity with UBE2L3[28]) suggested RNF213 could be a transthiolating E3 (i.e. covalently receives Ub from E2-Ub onto an active site cysteine), we used an activity-based probe (ABP) based on an E2-Ub conjugate, specific for this class of E3 mechanism[40]. The probe structurally mimics E2-Ub but is chemically modified to undergo irreversible covalent labeling of an E3's active site cysteine during transthiolation (Fig. 1g)[40]. Based on previous biochemical and proteomics data[28,32], we used the E2 UBE2L3 to generate the ABP and appended a biotin tag to enable detection (biotin-ABP). Consistent with the E2-Ub discharge assays, covalent ABP labeling of RNF213 was observed in the presence of ATPγS, but not with ADP or AMP (Fig. 1h). To detect E3 activation by ATP, elevated levels of nucleotide were required, which is presumably due its rate of hydrolysis by RNF213 being faster than that of the E3-ABP reaction[28] (Supplementary Figs. 1f–h). Consistent with this, the WB4 mutant underwent robust ABP labeling at ~100-fold reduced ATP levels indicative of sustained RNF213 activation and confirming that ATP hydrolysis counteracts ATP-dependent activation (Supplementary Figs. 1g, h). Together, these data support a model whereby 1) RNF213 undergoes transthiolation with E2-Ub, and 2) this fundamental step, which precedes substrate ubiquitination, is regulated by ATP binding.

In living cells adenylate homeostasis is maintained by adenylate kinase, which tightly couples ATP to ADP and AMP levels[41]. We used our

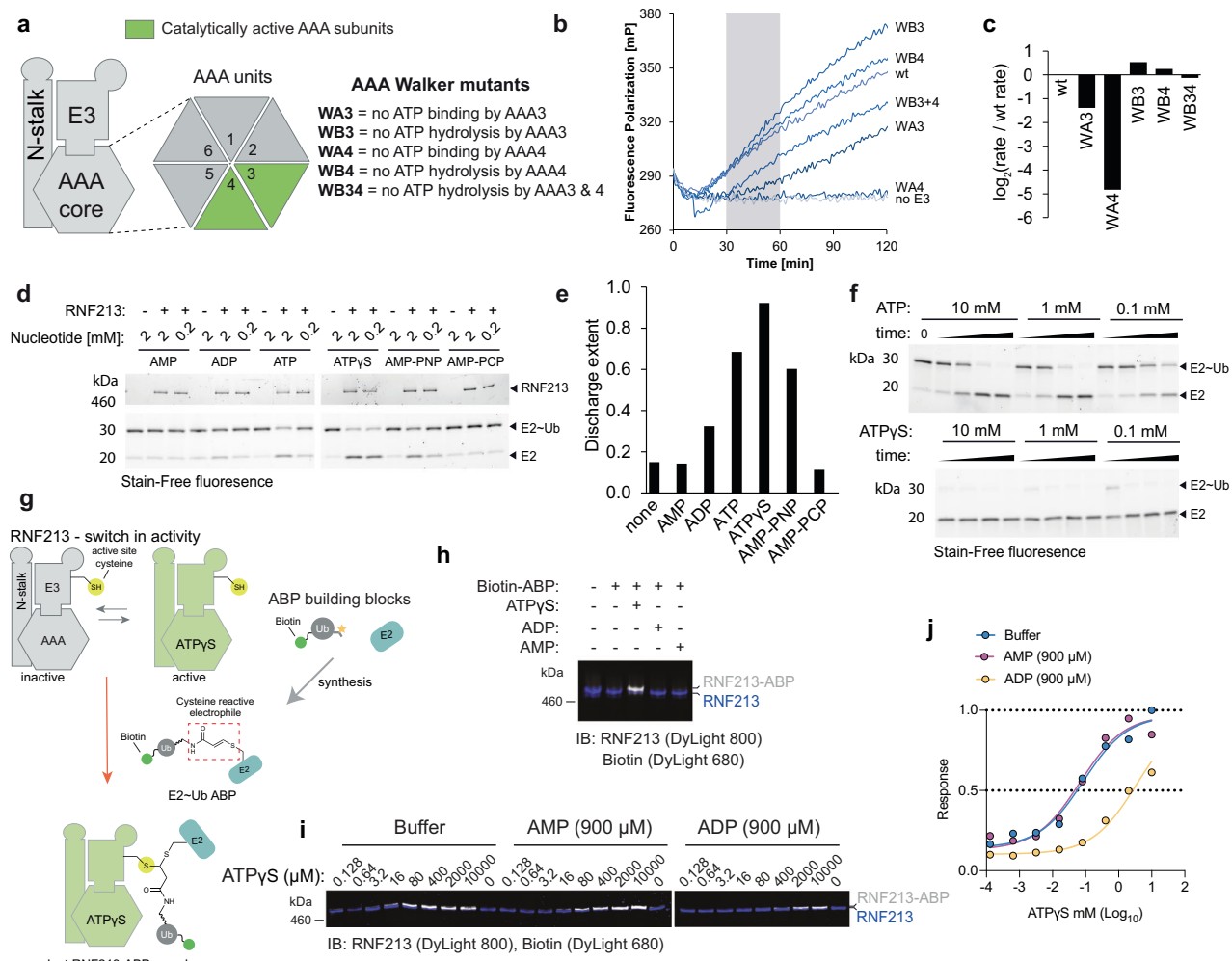

**Fig. 1 | Nucleotide-dependent regulation of RNF213 E3 activity. a** Schematic of RNF213 sub-modules (functional AAA3 and AAA4 in green) and mutations used to probe the role of the two functional AAA units in E3 ligase activity (WA3, K2387A; WA4, K2736A; WB3, E2449Q; WB4, E2806Q; WB34, E2449Q E2806Q; murine numbering). **b** Representative progress curves for the fluorescence polarization-based E3 activity assay. Reactions were carried out with wild type or mutant RNF213 in the presence of ubiquitin labeled sub-stoichiometrically with DyLight488. The time interval (30-60 min) used to determine rate constants is depicted in gray. **c** Rates of ubiquitin adduct formation (relative to wild type protein). **d** Discharge of Ub from purified UBE2L3-Ub discharge assay in the presence of RNF213 and various nucleotides. **e** Quantification of the discharge extent (free E2 / total E2). **f** Time

course of UBE2L3-Ub discharge assay, using different nucleotide concentrations. **g** Schematic of the nucleotide-dependent activation of RNF213 HECT-like E3 activity, required to form a covalent complex with an activity-based probe (ABP, wavy bond corresponds to a triazole-ethyl linker[40]). **h** Recombinant RNF213 was tested for transthiolation activity with a biotin-tagged ABP, in the presence or absence of 1 mM nucleotides indicated. Samples were resolved by SDS-PAGE and visualized by two-channel near-IR immunoblot using anti-RNF213 antibody (blue) and streptavidin (white). Panel representative of $n = 3$. **i** ABP analysis of ATPγS-dependent RNF213 activation in the presence of either ADP (900 μM) or AMP (900 μM). **j** Quantification of biotin signal determined in **i**. Mean values are plotted and non-linear fit curves shown ($n = 2$). Source data are provided as a Source Data file.

in vitro ABP assay to assess whether ADP or AMP could influence RNF213 activation by ATP. We initially determined an $EC_{50}$ for RNF213 activation by ATPγS of 85 ($\pm 32$) μM (Fig. 1i, j). In the presence of 900 μM AMP, the $EC_{50}$ was unaffected (67 ($\pm 28$) μM). However, with 900 μM ADP it increased ~40-fold to 3 ($\pm 2$) mM (Fig. 1i, j). These data suggest that ADP inhibits ATP-dependent activation. Moreover, as ATPγS was a stronger activator of RNF213 activity than ATP in E2-Ub discharge reactions (Fig. 1d–f), a higher $EC_{50}$ activation by ATP – approximating physiological millimolar ATP concentrations – would be anticipated inside cells.

## Cellular ATP levels regulate RNF213

We next asked whether RNF213 E3 activity is sensitive to adenosine nucleotides in cells. However, E3 ligase activity cannot be measured directly within live cells. To address this challenge, we developed a proteome-scale methodology enabling *in cellula* activity-based E3 profiling (Fig. 2a). Our E2-Ub ABPs can measure E3 transthiolation

activity in extracted proteomes[32], but whether they are functional within the subcellular environment remained unknown. To test this, we used capillary electroporation to enable non-destructive delivery of biotin-ABP into live cells[42]. After electroporation, cells were returned to the incubator, before being collected, washed and lysed, then incubated with streptavidin-coated resin to enrich the introduced biotin-ABP – including any E3 to which it has become covalently coupled. Importantly, addition of biotin-ABP to cells in the absence of electroporation did not result in accumulation of ABP inside cells, indicating that all enriched ABP was cell-internalised (Fig. 2b). Tryptic digest and mass spectrometry led to the identification of a series of E3s – including RNF213 – indicating that biotin-ABP electroporation allows us to profile the transthiolation activity of E3s inside living cells (Fig. 2c)[32]. To assess the cellular requirements of ATP, the AAA core, and the RZ domain for E3 activity, we generated an RNF213-null background in 293 cells (T-REx-293$^{RNF213KO}$) (Supplementary Fig. 2a) in which we stably

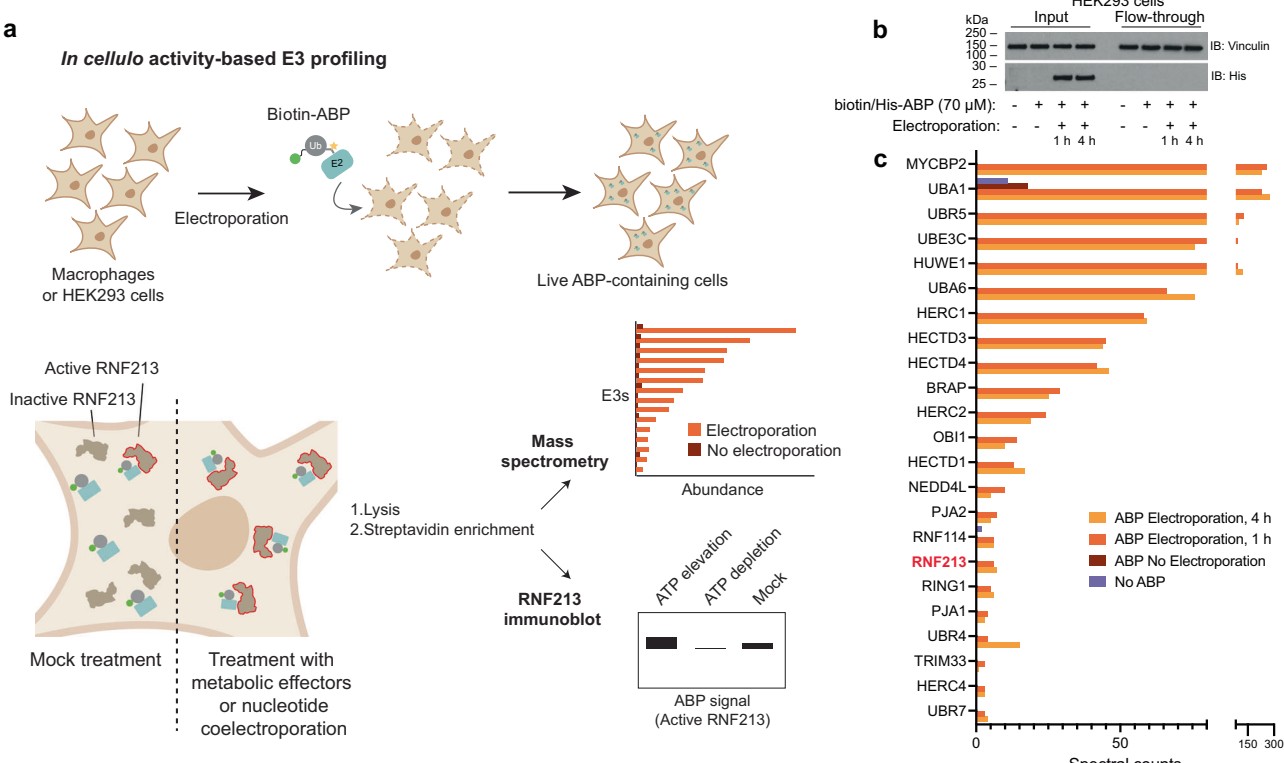

**Fig. 2 | *In cellula* activity-based E3 ligase profiling. a** *In cellula* activity-based E3 ligase profiling involving the delivery of a biotin-tagged E2-Ub ABP into live cells by electroporation. Labeled E3s can be subsequently enriched and analysed by mass spectrometry or Western blot. The abundance of a recovered E3 is a proxy for its transthiolation activity, allowing relative changes to be inferred. **b** Western blot demonstrating intracellular delivery of ABP by electroporation followed by subsequent incubation for the specified time. Following lysis, the ABP and its labeled proteins can be subsequently captured with streptavidin resin. Panel shown representative of *n = 3*. **c** Mass spectrometry analysis showing total spectral counts from E3s and the two E1 enzymes (UBA1 and UBA6) enriched by Biotin-ABP electroporated into 293T cells. Plotted proteins had >10 times more spectral counts 1 h post-electroporation than ABP (based on UBE2D3) treatment without electroporation. E3s were selected based on Pfam domain terms. The E3 coverage was similar whether ABP was incubated for 1 or 4 h post-electroporation. Source data are provided as a Source Data file.

expressed tetracycline-inducible human RNF213 variants. Due to the large relative size of RNF213, an electrophoretic shift upon ABP labeling – previously observed with smaller E3s[40] – could not be clearly resolved. To sidestep this problem, we enriched RNF213-ABP conjugates against streptavidin, as above, and probed Western blots for co-purified RNF213 to infer E3 activity (Fig. 2a). First, we determined the *in cellula* influence of ATP on RNF213 activity, decreasing ATP concentrations two-fold by inhibiting glycolysis with 2-deoxyglucose (2-DG)[43] (Fig. 3a). Electroporation of biotin-ABP into treated or untreated cells revealed a commensurate decrease in streptavidin-enriched RNF213 upon 2-DG treatment, while the streptavidin-enriched signal for the E1 enzyme UBA1, which we used as an internal control, was unaffected (Fig. 3b, Supplementary Fig. 2b)[32]. Conversely, co-electroporation of ATPγS robustly increased RNF213 recovery by streptavidin, indicating elevated RNF213-ABP labeling and thus RNF213 E3 activity (Fig. 3c). As our previous experiments showed that RNF213 E3 activity is linked to ATP sensing via AAA3 and AAA4 in vitro, the observed dynamic and bidirectional changes in cellular RNF213 are likely elicited by the same mechanism. These results also show that the human and murine RNF213 orthologues are regulated similarly by ATP/ADP ratio.

To confirm the role of the AAA3 and AAA4 ATPase sites on E3 activity in a cellular context, we delivered the biotin-ABP into T-REx-293[RNF213 KO] cells expressing RNF213 variants containing substitutions at AAA3 Walker A (K2426A, WA3), AAA3 Walker B (E2488Q, WB3) and AAA4 Walker A (K2775A, WA4); we were unable to recover stable WB4 variant expressing cells. RNF213 variants did not display equivalent expression patterns as assessed by immunofluorescence microscopy

(Supplementary Fig. 2c), meaning we were cautious to interpret relative basal E3 activities across variants. Instead, we compared each variant's sensitivity to co-electroporation of ATPγS. As before, and consistent with our in vitro measurements, ATPγS activated WT protein, whereas neither of the WA variants was activated by nucleotide (Fig. 3d, e), consistent with in vitro ubiquitination assays (Fig. 1b, c). In agreement with the WB4 mutant demonstrating sustained activation by ATP, the ABP signal for the WB3 mutant was significantly higher than WT (Fig. 3d, e). Collectively, this suggested that ATP-dependent E3 stimulation of RNF213 inside cells is also due to ATP binding, rather than hydrolysis. This also demonstrated that cellular ABP labeling correlated with RNF213 autoubiquitination and LPS ubiquitination activities observed in our biochemical assays.

### IFNs increase cellular ATP and uncouple RNF213 activity from expression

Armed with an approach for detecting subtle changes in E3 activities in cells, we asked whether we could address the now widely reported mechanism by which immune stimulation reveals RNF213 antimicrobial function. Thery et al. demonstrated that type I interferons (IFN), which induce broad antiviral states in cells, post-translationally license RNF213 antimicrobial behavior[21]. Similarly, type II IFNs licence RNF213-dependent bacterial and parasitic killing[23,24,26]. Given that RNF213 E3 activity and target ubiquitination are required for these broad antimicrobial effects[13,21,24,26], we hypothesized that IFNs could post-translationally activate the RNF213 E3 module via cellular energy levels. To explore this, we treated THP-1 cells with three subtypes of human IFN-I, the sole human IFN-II molecule (IFNγ), or a universal IFN-I

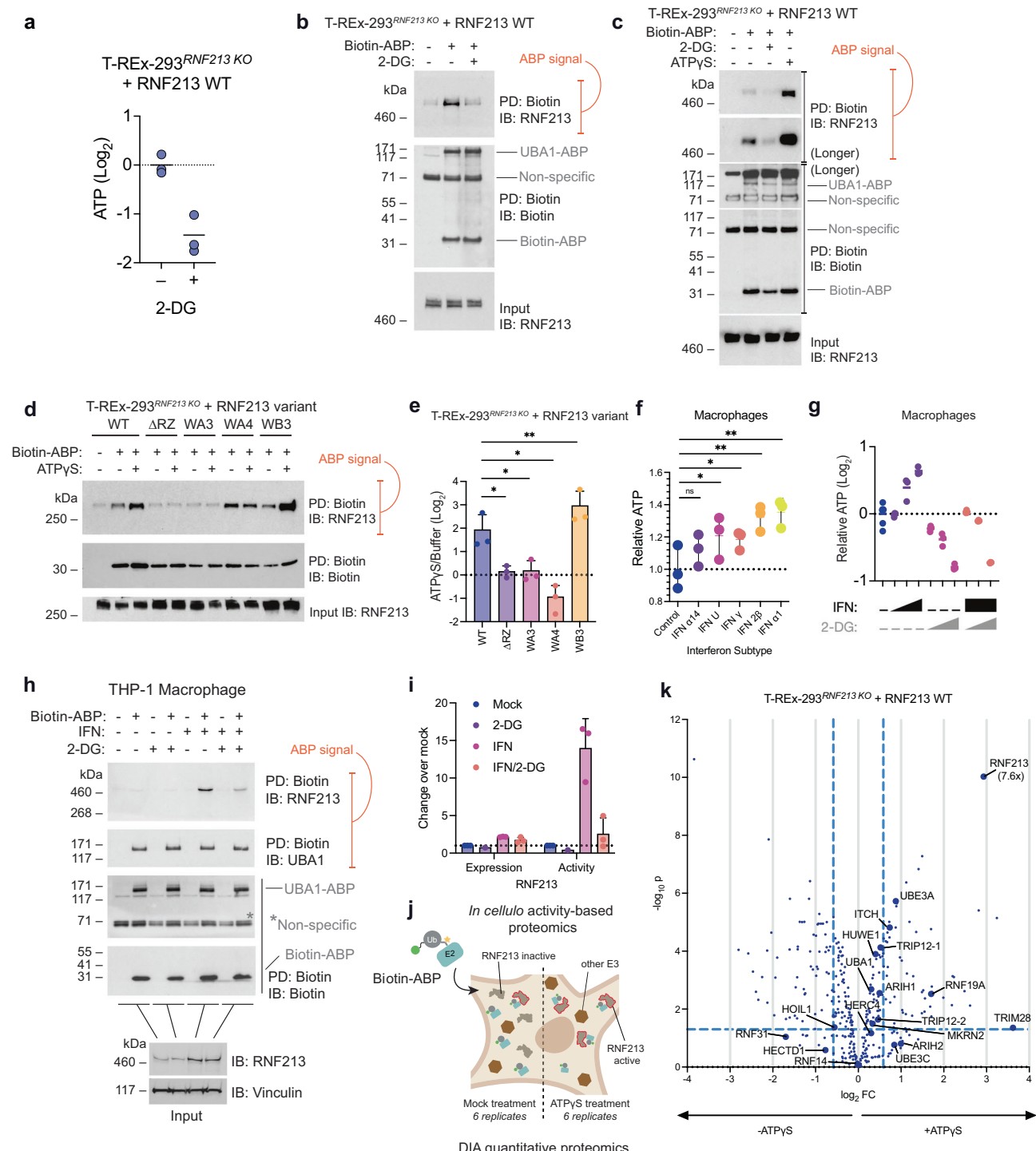

(IFN U), that stimulates diverse mammalian species, before measuring cellular ATP levels 16 h later. In all cases we detected a rise in cellular ATP upon IFN stimulation (Fig. 3f, g). To ask whether RNF213 E3 activity was sensitive to this metabolic shift, we treated THP-1 cells with or without IFN-I for 16 h, before electroporating cells with biotin-ABP. Profiling RNF213 activity following ABP enrichment revealed an elevation in RNF213 activity in cells treated with IFN-I (Fig. 3h). This increase was not accounted for by total RNF213 protein, which was modestly upregulated by IFN-I treatment in these cells, as previously reported[21] (Fig. 3h, i). As the increase in RNF213-ABP signal was more than could be accounted for by elevated protein expression, this suggested IFN-I treatment had post-translationally activated RNF213.

To test whether this was via the observed increase in ATP, we additionally treated cells with 2-DG for 3 h, which negated the IFN-I-induced ATP rise (Fig. 3g). Accordingly, 2-DG reversed the IFN-induced RNF213-ABP signal, but not the increase in RNF213 expression (Fig. 3h, i). Importantly, the residual activity observed in the +IFN, +2-DG condition reflects the concomitant increase in RNF213 expression by IFN (Fig. 3h). Thus, IFN uncouples RNF213 activity from expression, supporting a model whereby IFN reversibly activates the E3 via ATP. Consistent with this, the suppressive effect of 2-DG on RNF213 E3 activity was countered by the provision of ATPγS, further supporting a post-translational effect of IFN, and 2-DG, on RNF213 E3 activity (Supplementary Fig. 2d). Throughout these experiments in THP-1, and

**Fig. 3 | Assessment of RNF213 activation in live cells in response to changes in ATP levels. a** Treatment of RNF213 KO T-REx-293 cells stably overexpressing RNF213 (T-REx-293$^{RNF213\ KO}$ + RNF213) with 30 mM 2-deoxyglucose (2-DG) for 3 h results in a 2-fold reduction in ATP levels. Cellular ATP levels were determined by luminescent ATP assay. Data are presented as mean values ($n = 3$). **b** *In cellula* ABP analysis of T-REx-293$^{RNF213KO}$ + RNF213 cells that were mock, or 2-DG treated. RNF213 transthiolation activity is reduced in response to 2-DG treatment. UBA1, which also demonstrates transthiolation activity, is expressed at high levels and undergoes ABP labeling in a 2-DG insensitive manner. **c** Co-electroporation with ATPγS resulted in robust activation of RNF213 transthiolation activity. Panel representative of $n = 2$. **d** *In cellula* activity-based profiling of T-REx-293$^{RNF213KO}$ cells stably overexpressing the stated RNF213 variant in response to ATPγS. **e** Quantification (densitometry) of fold increase in ABP labeling comparing ATPγS- to buffer-electroporated control, from **d** (biological replicates $n = 3$). Two-way ANOVA, P values compare the variants to WT (left to right); 0.0371 (*), 0.0391 (*), 0.0133 (*), 0.0038 (**); data are presented as mean values +/- SD ($n = 3$). **f** ATP levels in monocyte-derived macrophage THP-1 cells 16 h after treatment with 100 U/mL IFN α14, Universal IFN (IFN-U), IFN γ, IFN 2β or IFN α1. Cellular ATP levels were determined by luminescent ATP assay. Data are presented as mean values +/- SD. One-way ANOVA, F value = 6.915, Degrees of Freedom = 10; adj. P values (left to right); 0.1175 (ns), 0.0423 (*), 0.0492 (*), 0.0040 (**), 0.0025 (**); error bars are SD ($n = 3$, representative experiment comprising technical replicates shown). **g** Monocyte-derived macrophage THP-1 cells treated with 2-DG (3, 10, or 30 mM) and IFN α1 (10, 100, or 1000 U/mL) for 16 h. Cellular ATP levels were determined by luminescent ATP assay. **h** *In cellula* transthiolation activity of endogenous RNF213 in IFN-stimulated macrophages that were mock or 2-DG treated. **i** Quantification (densitometry) of fold increase in Western band intensity across conditions, relative to mock treatment, from **h** (biological replicates $n = 3$; 2-DG-alone was included in 1 replicate as the signal was at or below detection threshold). Data are presented as mean values +/- SD. **j** Schematic illustrating how combining *in cellula* activity-based E3 ligase profiling with data-independent acquisition mass spectrometry (DIA-MS) allows the activity changes of a cohort of E3s to be simultaneously quantified upon cellular delivery of ATPγS. **k** Relative changes in ABP (based on UBE2L3) signal upon ATPγS delivery into T-REx-293$^{RNF213\ KO}$ + RNF213 determined by *in cellula* activity-based E3 profiling and DIA-MS. Quantification was performed by analysing DIA spectra ($n = 6$) with DiaNN (v1.8.1) operating in library-free mode. Protein regulation was assessed using two-tailed differential expression analysis carried out in LIMMA using an empirical Bayes-moderated linear model and P values were corrected using Benjamini−Hochberg multiple hypothesis correction. The hashed vertical lines correspond to fold changes of 1/1.5 and 1.5, whereas the horizontal hashed line corresponds to an adjusted *P*-value cut-off of 0.05. Enlarged data points correspond to HECT/HECT-like E3s and the ubiquitin E1 UBA1. Two TRIP12 isoforms were detected and designated TRIP12-1 and TRIP12-2. Source data are provided as a Source Data file.

as in HEK293 cells, the ABP signal for UBA1 was unaffected by any treatment (Fig. 3h), arguing against non-specific effects of IFN or 2-DG on ABP reactivity.

Thery et al. also demonstrated that IFN induces RNF213 oligomerisation on lipid droplets, in a manner requiring conjugated ISG15. A prediction of this model was that higher order assembly activated the RNF213 E3 and thus antimicrobial function[21]. To test this using our assay, we inhibited ISG15 conjugation by stable depletion of Ube1L from THP-1 cells (Supplementary Fig. 2e), which, as expected, impaired ISG15 conjugation, but not the levels of unmodified ISG15 (Supplementary Fig. 2f). ABP electroporation revealed reduced – but not inhibited – RNF213 E3 stimulation by IFN following Ube1L depletion (Supplementary Fig. 2g), supporting a role for ISG15 in RNF213 E3 activation. To explore whether oligomerisation was required for IFN-induced E3 activation, we sought microscopic evidence for RNF213 lipid droplet association, however we were unable to visualize RNF213 on lipid droplets in differentiated THP-1 cells, with or without IFN stimulation. Nor did we observe RNF213 oligomerisation in the presence or absence or ATPγS in vitro, as assessed by size exclusion chromatography of recombinant RNF213 in the presence or absence of ATPγS (Supplementary Fig. 2h). We also did not observe ISG15 modification of RNF213 immunoprecipitated from IFN-treated THP-1. Interestingly, ISG15 also regulates cellular ATP production by modification of glycolytic and mitochondrial enzymes[44–46] and we observed significantly reduced ATP levels in Ube1L-depleted THP-1 cells (Supplementary Fig. 2i). Moreover, while IFN did increase ATP levels in both control and Ube1L-depleted cells, in the latter these did not exceed those of untreated control cells (Supplementary Fig. 2j), suggesting reduced E3 activation upon Ube1L depletion could also stem from a failure to surpass an ATP activation threshold. While we cannot exclude a structural role for ISG15 in RNF213 E3 activation, our current data support a model of ATP-dependent RNF213 activation.

Also consistent with our biochemical observation that ADP antagonizes ATPγS-dependent activation (Fig. 1j), co-electroporation of biotin-ABP and ADP inhibited cellular RNF213 E3 activity (Supplementary Fig. 2k). Contrary to our biochemical assay, AMP also inhibited RNF213 activity. We presume this is by the presence of cellular adenylate kinase which salvages AMP by its ATP-dependent conversion into ADP (AMP + ATP -> 2ADP), which we showed is inhibitory[41]. Combined, these experiments suggest a mechanism whereby IFN regulates ATP levels and, in addition to increasing RNF213 expression,

augments RNF213 E3 activity post-translationally, which we can detect using our live-cell E3 profiling methodology.

**ATP-E3 regulation is not a general phenomenon**

Of the other >600 annotated E3s in the human genome, none are yet known to couple direct nucleotide binding to E3 activity, suggesting RNF213 demonstrates a novel regulatory paradigm. To test this notion empirically, we agnostically assessed the effect of ATPγS on the transthiolating activity of a cohort of E3s. This was achieved by *in cellula* E3 activity profiling of T-REx-293$^{RNF213\ KO}$ cells expressing WT RNF213 in the presence or absence of electroporated ATPγS, followed by quantitative data-independent mass spectrometry (DIA-MS) (Fig. 3j). As above, we found that the ABP signal for RNF213, representing the active E3, was robustly enriched by >7-fold, upon ATPγS electroporation (Fig. 3k). Of the other transthiolating E3s detected (5 RBR and 8 HECT E3s), only the HECT E3s ITCH and UBE3A, and the RBR E3 RNF19A, were enriched above the significance threshold (Fig. 3k). Interestingly, both UBE3A and RNF19A interact with components of the AAA-ATPase 19S proteasome regulatory particle raising the potential for direct coupling of ligase and proteasome activity[47–49]. Similarly, the AAA ATPase VPS4A is an ITCH substrate[50]. This suggests the activities of these E3s are influenced indirectly by modulation of the ATPase activity of their partner proteins, which presumably would be perturbed by ATPγS. Taken together, intracellular E3 activity profiling supports our model that RNF213 has an intrinsic ability to be robustly modulated by subcellular changes in nucleotide concentrations, by coupling ATP binding to E3 activity. The uniqueness of RNF213 in this system further rules out the possibility that altered nucleotide concentration was having a pleiotropic effect on the ubiquitin system. Such a mechanism has not been observed in an E3 before, encouraging us to better define the transthiolation mechanism of this unusual 600 kDa enzyme.

**Cryo-EM structures of the RNF213-E2 transfer complex**

We next determined the cryo-EM structure of the RNF213-Ub-UBE2L3 transfer complex, using the covalently linked RNF213-E2-Ub ABP adduct[32,51] (Fig. 4a). The 3.5 Å structure (Fig. 4b, c, Supplementary Fig. 3), revealed that UBE2L3 is bound via a well-defined interface formed between the C-terminal domain (CTD) of RNF213 and the L1/L2 loops and H1 helix of UBE2L3 (Fig. 4d). Despite induction of ABP-E3 complex formation with ATPγS, no clear nucleotide density was evident in sites AAA3 or AAA4. The CTD (residues 4930-5148) forms a

helical-rich domain centrally positioned in the RNF213 scaffold. It resembles no structurally characterized domain and is a novel E2 binding fold. CTD residues Phe5093, Trp5097, and Tyr5106 are oriented to undergo hydrophobic interactions with Phe63 and Pro97

located on the UBE2L3 L1 and L2 loops, respectively. This contact is putatively strengthened by nearby electrostatic interactions formed between CTD Glu5108 and Arg6 in the UBE2L3 H1 helix, and CTD Asp5101 and Lys96 in the UBE2L3 L2 loop. In E2s, the same L1/L2 loops

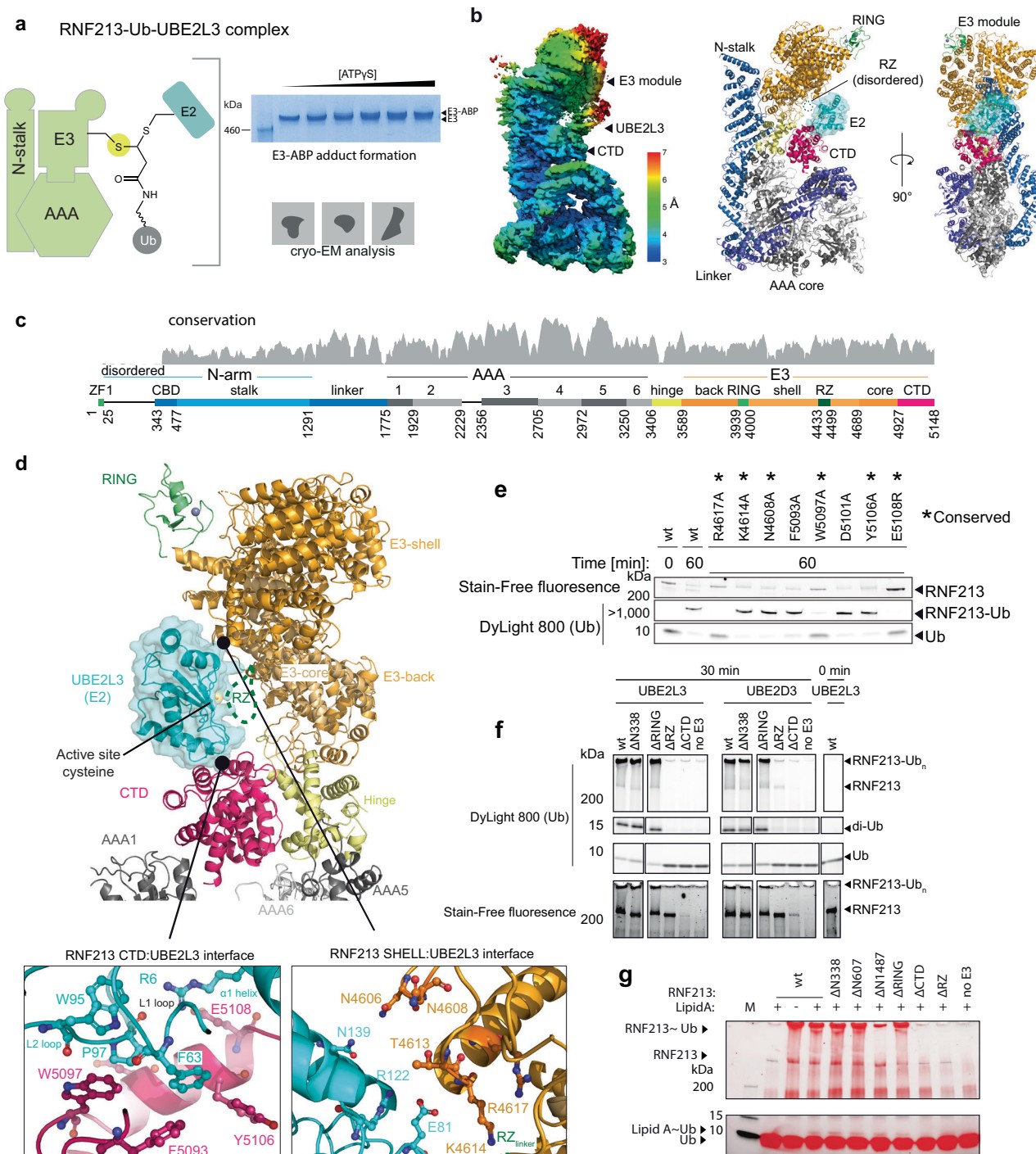

**Fig. 4 | Cryo-EM structure of the RNF213-UBE2L3 transthiolation complex.**
**a** Schematic of the covalent RNF213-UBE2L3-Ub complex, mimicking the transient E2-E3 transfer (transthiolation) intermediate. Inset illustrates complex formation by SDS-PAGE, demonstrating >90% labeling of full-length murine RNF213 (1.5 μM) at the highest ABP concentration (68 μM) in the presence of ATPγS (5 mM). Panel representative of $n = 3$. **b** Left, cryo-EM density of RNF213-UBE2L3-Ub complex at 3.5 Å resolution. A distinctive density appears between the E3 shell and the CTD domain of RNF213, into which UBE2L3 can be unambiguously docked. Right, ribbon representation of the RNF213-UBE2L3 model. Details about the cryo-EM reconstruction are given in Supplementary Fig. 4. **c** Domain architecture of RNF213.

**d** Close-up view of the E3 module with bound UBE2L3 (color coded according to E3 portions, and UBE2L3 shown in light blue). The two interfaces with CTD and E3-shell are shown enlarged in insets. Comparison with E2 interfaces from other E3 enzymes is given in Supplementary Fig. 5. **e** Autoubiquitination assay used to validate the structurally determined UBE2L3 binding site. **f** Autoubiquitination assay with various deletion mutants of RNF213 when partnered with UBE2L3 or UBE2D3. Δ338 assesses the role of the N-terminal residues that were disordered in the cryo-EM structure. **g** Ubiquitination and lipid ubiquitination assays with RNF213 deletion mutants (auto-ubiquitination, top panel; lipid A ubiquitination, lower panel). Source data are provided as a Source Data file.

and α1 helix mediate binding to RING, HECT, RBR, and RCR E3 ligases, as well as bacterial effector E3s[51–55] (Fig. 4d, Supplementary Fig. 4a). A secondary putative interaction site with the E2, formed by a protruding portion of the RNF213 E3-shell domain (residues 4605-4618), may facilitate binding and orientation of the E2 enzyme (Fig. 4d). Mutation of R4617, a proximal residue that may stabilize the secondary interaction site, disrupted E3 activity (Fig. 4e). However, this could also be due to disrupted RZ domain dynamics, given R4617's proximity to loop regions that connect it to the E3 shell (Fig. 4d). We could not detect direct binding of the isolated purified CTD domain (residues 4986-4207) to UBE2L3, nor nucleotide-dependent binding of full-length RNF213 to E2 or E2-Ub, indicative of transthiolating activity having a high Michaelis constant ($K_m$) that is driven by catalytic turnover ($k_{cat}$) (Supplementary Figs. 4b–f).

To validate the RNF213-E2 structure we performed a mutational analysis of key interface residues (Fig. 4e). CTD mutations W5097A and E5108R resulted in loss of E3 activity comparable to the deletion of the entire CTD, confirming the location of the E2 binding site on RNF213 and consistent with in vivo data demonstrating that deletion of the 20 C-terminal residues, likely to destabilize the entire CTD, prevents RNF213 from ubiquitinating bacterial LPS[13]. The dependence on the CTD domain remained regardless of substrate (RNF213 or Lipid A) and substrate specificity was E2 independent (UBE2L3 or UBE2D3), indicating that E2 recruitment by the CTD precedes all downstream ubiquitination events (Fig. 4f, g, Supplementary Fig. 1a). The RNF213 RING domain is dispensable for autoubiquitination and LPS ubiquitination[13,28]. Nevertheless, we asked whether the isolated domain demonstrated catalytic activity. We tested for its ability to stimulate discharge from the aminolysis proficient E2 UBE2D3 to free lysine but no activity was detectable (Supplementary Fig. 5a).

In the RNF213-E2 complex, the E2 active-site residue Cys86 is oriented towards the center of the E3 pocket formed by RNF213, facing a low-contour-level density that presumably originates from the RZ domain (residues 4433-4499) (Fig. 4d)[13]. MS analysis confirmed that the RNF213-E2-Ub ABP sample contains Ub, despite not having a well-defined position in the complex that could be structurally resolved, which might be reflective of the Ub and RZ domain being inherently dynamic, and/or the RZ domain having a degree of structural disorder (Supplementary Figs. 5b, c)[56–59]. Indeed, it is not uncommon that the ubiquitin-transfer step of E3 enzymes cannot be captured by structural data; the catalytic cysteine domain (RING2) of the E3 Parkin was highly dynamic in the activated step, necessitating its deletion to facilitate structure determination[60]. As the majority of E3s are thought to target a variety of substrates, the active site organization and Ub transfer process may rely on inherently mobile structural elements, further exemplified by the large family of SCF ubiquitin ligases[61].

As nucleotides were absent from sites AAA3 and AAA4 in our RNF213-E2-Ub structure, we postulated that RNF213 might have reverted to an inactive conformation during sample processing, rendering our structure a covalently linked "footprint" of the activated transthiolation intermediate. To explore this possibility, we generated a second cryo-EM structure where the ATPγS-stimulated ABP labeling reaction was applied directly to cryo-EM grids (Supplementary Fig. 6 and Table 1). Modest structural changes were evident (RMSD = 2.66 Å) in the AAA core and new density at the AAA3 and AAA4 sites allowed modeling of ADP (Supplementary Figs. 7a–d). However, there remained no clear density for the RZ domain nor Ub and no apparent basis for activation of transthiolation. To enforce an ATP bound state, reasoning this might capture the activated conformation, we determined a structure of the RNF213 WB34 double mutant in the presence of ATP (Supplementary Fig. 8 and Table 1). Structural changes relative to the original ABP-complex were less pronounced (RMSD = 1.57 Å). Although ATP could be modeled into AAA4, AAA3 was empty and once again, there was no clear RZ/Ub density and no clear basis for elevated transthiolating activity (Supplementary Fig. 9a–d). These findings

**Table 1 | Cryo-EM data collection, refinement, and validation statistics**

| | RNF213-ABP (EMDB: EMD-12931) (PDB: 7OIK) | RNF213-ABP ATPgS (EMDB: EMD-####52571) (PDB: ####9I1J) | RNF213-WB3/WB4 + ATP (EMDB: EMD-####52570) (PDB: (9I1I) |
|---|---|---|---|
| **Data collection and processing** | | | |
| Voltage (kV) | 300 | 300 | 300 |
| Electron exposure (e⁻/Å²) | 32.5 | 50 | 30.8 |
| Defocus range (µm) | –0.8 to –2.5 | –0.8 to –2.5 | –1 to –2 |
| Pixel size (Å) | 1.072 | 1.178 | 1.06 |
| Symmetry imposed | C1 | C1 | C1 |
| Initial particle images (no.) | | | |
| Final particle images (no.) | 98 000 | 31 607 | 33 242 |
| Map resolution (Å) | 3.5 | 3.8 | 4.5 |
| FSC threshold | 0.143 | 0.143 | 0.143 |
| **Refinement** | | | |
| Initial model used (PDB code) | 6TAX, 4Q5E | 6TAX, 4Q5E | 6TAX |
| Model resolution (Å) | 4.2 | 7.3 | 7.3 |
| FSC threshold | 0.5 | 0.5 | 0.5 |
| Model composition | | | |
| Non-hydrogen atoms | 36779 | 36793 | 36856 |
| Protein residues | 4579 | 4576 | 4585 |
| Ligands | ATP, Zn (2), Mg | ATP, ADP (2), Zn (2), Mg | ATP (2), Zn (2), Mg |
| *B* factors (Å²) | | | |
| Protein | 256 | 244 | 248 |
| Ligand | 195 | 180 | 184 |
| R.m.s. deviations | | | |
| Bond lengths (Å) | 0.006 | 0.003 | 0.003 |
| Bond angles (°) | 1.32 | 0.73 | 0.73 |
| Validation | | | |
| MolProbity score | 1.8 | 2.31 | 2.32 |
| Clashscore | 5.4 | 13.6 | 16.8 |
| Poor rotamers (%) | 0.0 | 1.88 | 1.46 |
| Ramachandran plot | | | |
| Favored (%) | 90.9 | 92.8 | 92.6 |
| Allowed (%) | 8.9 | 7.18 | 7.42 |
| Disallowed (%) | 0.2 | 0.04 | 0.02 |

suggest the ATP activated state is highly transient or involves very modest structural changes that allosterically reposition the RZ domain.

## RNF213 undergoes E2-E3 ubiquitin transfer via its RZ domain

The labeling of RNF213 with an ABP for transthiolating E3s, and its autoubiquitination activity with UBE2L3, indicated that it depends on an active-site cysteine nucleophile. To identify this key residue, we performed cross-linking mass spectrometry (XL-MS) analysis of the

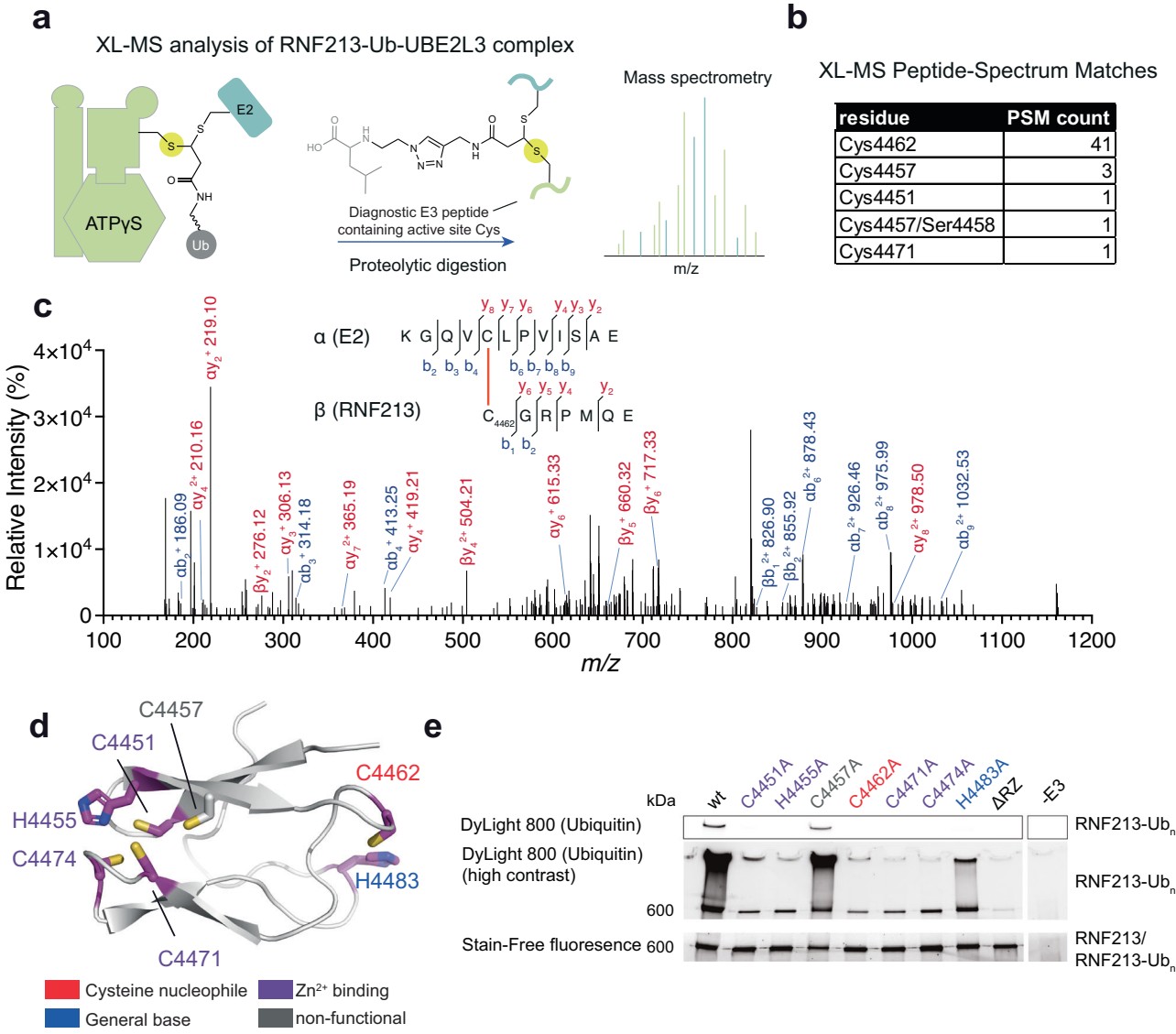

**Fig. 5 | ABP mapping of the E3 cysteine nucleophile in the RZ domain. a** Cross-linking mass spectrometry (XL-MS) analysis of the full-length RNF213-ABP complex. The residue labeled by the ABP corresponds to the active site cysteine nucleophile (yellow circle), which was identified by XL-MS. **b** Cross-link peptide-spectrum matches (PSMs) that have an e-value lower than $1 \times 10^{-4}$ with a false discovery rate (FDR) of <1% are tabulated. **c** Representative $MS^2$ spectrum of a crosslinked peptide corresponding to Cys4462 in murine RNF213. The spectrum is for a $z = 4^+$ precursor ion, observed precursor mass = 2369.186 Da; theoretical crosslinked peptide mass = 2369.189 Da. **d** Alphafold2 model of the murine RNF213 RZ domain. Consistent with ABP crosslinking analysis, UV-VIS absorbance spectroscopy, and activity assays, Cys4451, His4455, Cys4471, and Cys4474 coordinate a single metal ion. Cys4462 is the active site nucleophile essential for RNF213 transthiolation activity. His4483 is suitably positioned to serve as a general base that facilitates deprotonation of substrate nucleophiles. **e** Autoubiquitination assay with UBE2L3 using the point mutants of cysteine and histidine residues within the RZ domain. Source data are provided as a Source Data file.

covalent RNF213-E2-Ub ABP complex[32,40] (Fig. 5a). This experiment strongly converged on Cys4462 as the active site residue to which the E2-Ub ABP is coupled, which resides within the RZ domain in mouse and human RNF213 (Fig. 5b, c). Structural modeling of the RZ domain using Alphafold2[62] revealed that Cys4462 does not contribute to zinc ion coordination but is located in a remote loop segment (Fig. 5d). Deletion of either the entire RZ domain, or mutation of Cys4462, led to complete loss of in vitro autoubiquitination and Lipid A activity (Figs. 4g, 5e). Similarly, in cells, no ABP signal, and thus no E3 activity, was observed for RNF213 ΔRZ, consistent with this domain harboring the site of E3 activity in cells (Fig. 3d, e)[13]. Attempts to trap a catalytic E3-Ub intermediate with a C4462S mutant were unsuccessful because the serine mutant retained autoubiquitination activity, as observed with other E3s[32], such that the potential intermediate was undetectable against the background signal (Supplementary Fig. 9e).

We validated the in silico fold prediction of the RNF213 RZ domain using ultraviolet-visible spectroscopy (UV-VIS), confirming that it has a divalent transition metal-binding fold[63,64], with a preference for $Zn^{2+}$ (Supplementary Figs. 10a, b) and requires the predicted zinc-coordinating residues Cys4451, His4455, Cys4471, and Cys4474 (Fig. 5d, Supplementary Figs. 10c–e). Mutation of any of these residues ablated autoubiquitination, indicating that the structural integrity of the RZ domain through metal binding is required for RNF213 E3 activity (Fig. 5e). A remaining cysteine in the RZ domain is Cys4457, but this is unlikely to have a functional role because it is neither conserved nor required for metal binding or E3 ligase activity (Fig. 5e, Supplementary Figs. 10c–e, Supplementary Fig. 7). However, His4483 is located adjacent to Cys4462 in the structural model and is highly conserved (Fig. 5d, Supplementary Fig. 11a). Mutating this histidine to alanine led to lower (<10x) autoubiquitination activity (Fig. 5e),

suggesting His4483 functions as a general base in a catalytic Cys/His dyad. In conclusion, our data show that the RNF213 uses the Cys4462/His4483 dyad as a catalytic motif, located on adjacent loops of the RZ domain, for E3 activity.

## Discussion

Here, we provide a functional characterization of the Ub transfer mechanism of the giant E3 RNF213. Our data suggest the following sequence of events: IFNs, released during infections, increase cellular ATP levels; RNF213 senses changes in ATP/ADP ratios via its AAA core, which regulates its atypical transthiolating E3 module; the E2-Ub docks onto RNF213 via the CTD, while the RZ domain provides the catalytic cysteine nucleophile required for transthiolation with E2-Ub; ubiquitin is transferred to Cys4462, forming the reactive acyl donor tasked with substrate modification; additional contacts between the E2 and the E3 shell, and Ub and the RZ domain, may optimize Ub transfer. The ADP generated by ATP hydrolysis, which we show is inhibitory, may provide a mechanism to attenuate RNF213 transthiolation, thereby keeping RNF213 activity strictly localized to subcellular regions of elevated ATP/ADP ratio (Fig. 6). How this allosteric mechanism is achieved is not yet clear, nor how diverse substrates are recognized and positioned for ubiquitination. We observe no notable structural differences between the RNF213 in our RNF213-ABP complex and that of isolated protein (root mean square deviation (RMSD) = 0.76 Å)[28]. Although we do not elucidate the structural detail of the active site itself, our structure suggests ATP binding induces either long-range allosteric repositioning of the RZ domain, or conformational fine-tuning of the E2-docking CTD, to enable transthiolation from the bound E2-Ub conjugate to the E3 active site. The structural rearrangements that facilitate activation might also be highly transient and recalcitrant to our cryo-EM workflow. Superposition of an RNF213 AlphaFold3 prediction results in steric clash between the RZ domain and the E2 density in our cryo-EM structure (Supplementary Fig. 11b), implying that the RZ domain is mobile, consistent with lack of clear density in our cryo-EM maps, and that E3 activation requires its reorientation. Nonetheless, our approach connected mechanistically the AAA and E3 modules of RNF213 and has added support to the use of ABPs in resolving fundamental active site chemistries of complex E3 ubiquitin ligases[51,65].

The timely activation of E3s is essential for cellular homeostasis and perturbations in this process result in diseases such as autoimmunity, cancer and neurodegeneration. However, the regulatory mechanisms for the vast majority of E3s remain unknown. With the emergence of E3s as compelling therapeutic targets, technologies are needed to uncover therapeutic opportunities and address drug discovery challenges, such as assessment of target engagement and drug selectivity. We anticipate that our methodology for proteome-scale assessment of E3 activity within live cells will alleviate some of these hurdles and have utility traversing cellular function, biomarker discovery, therapeutic target identification and drug discovery.

the other classes of E3, the RCR E3, and at least one RBR E3, contain zinc-binding transthiolation domains that also mediate modification of amino and hydroxy substrates[32,66]. For these E3 subtypes, the cysteine responsible for substrate modification is in a structured region of their respective zinc-binding domains and in immediate proximity of a free histidine, exemplified by MYCBP2. The histidine acts as a general base that deprotonates the substrate nucleophile and its mutation abolishes activity[32,67]. In contrast to MYCBP2, RNF213 is able to perform both O- (lipid A) and N- (lysine) linked ubiquitination, including the formation of polyubiquitin chains[28], underscoring the differences between these two E3s. However, our identification of the Cys4462/His4483 catalytic dyad in the RZ domain suggests a conserved catalytic strategy between RNF213 and MYCBP2 and raises the possibility of uncovering related E3 ligases that share this motif, similarly to the annotation of RING-type E3s. Indeed, Blast searches revealed that the same Zn-finger, including the Cys/His dyad, is present in the C-terminal region of human ZNFX1 (Supplementary Fig. 11a), an RNA helicase (and thus also an ATPase) similarly involved in the immune response against viral[68] and bacterial[69] infection. Consistent with ZNFX1 belonging to the same E3 subtype as RNF213, ZNFX1 is also enriched in HECT-like E3 activity-based proteomic data, prior to filtering against domains with established E3 activity[32]. Whether ZNFX1 demonstrates E3 ligase activity and if its ATPase activity is coupled to this, remains to be seen. The RNF213/ZNFX1 RZ domain[13], and the herein elucidated Ub transfer mechanism, could more broadly carry out the priming ubiquitination event to mark microbial invaders for autophagic clearance and stimulate IFN signaling, as part of cell-autonomous immunity.

The finding that RNF213 E3 activity is tuned to sense cellular ATP levels during an interferon response provides important clues to its broad antimicrobial sensitivity[13,21–24]. Moreover, certain RNF213-sensitive bacteria, including *Salmonella*, which secretes ATP into its surrounding milleu[70,71], and *Chlamydia*, which causes a spike in the host cell ATP/ADP ratio[72], could activate RNF213 in proximity to intracellular, metabolically active bacterial cells. RNA and DNA viruses are also reported to increase intracellular ATP in mammalian cells[73,74]. Beyond eukaryotes, a conserved bacterial anti-phage ATP nucleosidase was recently described[75], while the intracellular detection of nucleotides by proteins like Stimulator of Interferon Genes (STING), is a recurring theme for immune systems in eukaryotes and prokaryotes, in which these systems first arose[76]. Thus, we speculate that ATP could be an ancient correlate of infection, and its detection and/or hydrolysis a conserved immune strategy.

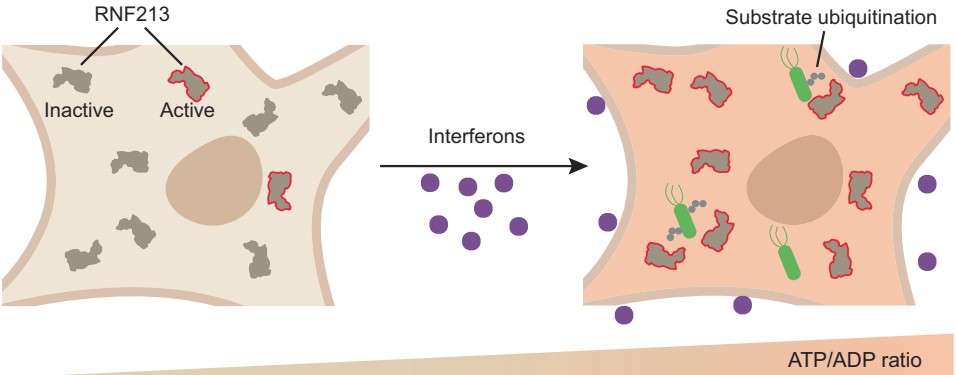

**Fig. 6 | Proposed model for RNF213 E3 activation upon sensing increased ATP/ADP ratio.** RNF213 E3 activity is tightly coupled to the cellular ATP/ADP ratio. Interferon stimulation causes an increase in ATP levels, which induces RNF213 E3 activity and substrate ubiquitination.

Supporting this, ATP is a well-described DAMP in the extracellular space, where through binding to cell surface purinergic receptors it influences inflammatory responses[4,77]. Although an intracellular PAMP/DAMP-like role for ATP has not been described per se, intracellular ATP as a correlate of infection is not without precedent. Nucleotide-binding leucine-rich repeat (NLR) proteins trigger inflammatory pyroptosis upon infection[78]; NLR pyrin domain-containing protein 3 (NLRP3) couples ATP binding to its AAA+ ATPase with higher-order assembly and inflammatory caspase activation[79,80]. Structures of NLRP3 in active or inactivate conformations suggest that ATP promotes higher-order assembly and activation, while ADP favors the inactive conformation[81,82]. By inducing inflammation, NLRP3 also has broad antimicrobial effects[83]. Intriguingly, mirroring its requirement for the activation of transthiolation, ATP binding induces RNF213 self-association in cells[35]. Although activation of transthiolation through oligomerization is a compelling hypothesis, we could neither observe ATP-dependent oligomerisation structurally nor biochemically, hence this may be reserved for membrane targeting[84]. Nonetheless, ATP as an intracellular cue for immune responses across the tree of life is now well-documented.

The link between IFNs and cellular nucleotides was reported as long ago as 1979[85] and more recently in Abt et al.[86]. In keeping with our observations in a myeloid cell line, IFNs were shown to enhance glycolysis and increase ATP in plasmacytoid dendritic cells[87]. Moreover, IFN-enhanced glucose uptake, glycolysis and ATP production, contributed to restriction of an RNA virus in mouse fibroblasts[88]. That we observe a similar response across distinct type I IFN subtypes and the only type II IFN is suggestive of a general innate immune phenomenon. While the mechanism connecting IFN to ATP requires elucidation, our data suggest that ISG15 conjugation, induced by IFN signaling, may be involved, although we note that cells deficient in ISG15 conjugation were generally reduced for ATP, and not just during an immune response. Nonetheless, our data suggest that in addition to supplying the energy to mount an immune response, IFN-induced ATP provides post-translational control of antimicrobial ATPases whose expression, like RNF213, is not wholly dependent on IFN. This mechanism could contribute to a rapid host response to diverse pathogen infections.

## Methods

Protein sequence analysis, cloning of mouse RNF213 expression constructs, expression and purification of mouse RNF213 protein, preparation of fluorescently labeled ubiquitin, the RNF213 SDS-PAGE-based auto-ubiquitination assay, cryo-EM analyses, and model building were performed as described before[28]. Discrepancies and new methods are outlined below.

### E2 discharge assay

The E2 discharge assay was performed using a purified UBE2L3-Ub thioester intermediate. UBE2L3-Ub was produced by preparing a reaction mixture consisting of 2.5 μM human UBA1, 170 μM human UBE2L3, 300 μM bovine ubiquitin, and 2 mM ATP, and incubating it at 37 °C for 20 min. The reaction mix was optionally supplemented with a 1:200 molar ratio of fluorescently labeled human ubiquitin with an N-terminal overhang containing a cysteine residue, labeled either with DyLight 488 Maleimide or DyLight 800 Maleimide (Thermo Fisher Scientific). The reaction was performed in the ubiquitination buffer (25 mM HEPES, 150 mM NaCl, 10 mM MgCl$_2$, 2 mM TCEP, pH 8.0 @ 25 °C). Afterwards, the reaction mixture was supplemented with 20 mM imidazole and incubated for 1-5 minutes over Ni-NTA agarose beads (Qiagen). Flowthrough was then collected after spinning the suspension down in a cellulase acetate spin column (Thermo Scientific #69702), separating the His-tagged UBA1 from the other components. Finally, the flowthrough was purified by SEC using a Superdex 75 Increase 3.2/300 column (Cytivia), using the ubiquitination buffer as the elution buffer, separating UBE2L3-Ub from remaining UBA1, free Ub, and ATP. UBE2L3-Ub produced in this way had a concentration between 40 and 80 μM and contained only a minor fraction (<10%) of free UBE2L3 as a contaminant. Aliquots were flash-frozen and stored at −70 °C. The purified adduct was stable on ice for at least 1 week, and for at least 1 year at −70 °C, and had remained fully stable for at least 32 freeze-thaw cycles. The discharge assay components were as follows: 0.025 μM E3, 20 μM E2-Ub, and 1 mM nucleotide (AMP, ADP, ATP, ATPγS, AMP-PNP, or AMP-PCP).

### Fluorescence polarization-based ubiquitination assays

The E3 ligase activity of RNF213 variants was measured using a fluorescence polarization (FP) based assay using 25 mM HEPES, 150 mM NaCl, 10 mM MgCl$_2$, 2 mM TCEP (VWR #97064-848), pH 8.0 as the reaction buffer, supplemented with 10 μM of BSA monomer (Sigma Aldrich #A1900). The typical concentrations of the components in the standard assay were as follows: 20 μM Ub, 0.5 μM E1, 8 μM E2, 0.05 μM E3. Ubiquitin source (from bovine erythrocytes) was always spiked with a 1:200 molar ratio of Ubiquitin-DyLight488 (from E. coli). Human E2 variants UBE2L3 and UBE2D3 were used, purified from E. coli. For E1, human UBA1 purified from E. coli was used. A 2X master mix was typically prepared with all the components apart from the screened component. 2 μL of the screened component at 2X final concentration was mixed with the master mix, yielding a 4 μL reaction mixture. 3.5 μL of the resultant mixture was transferred to a 1536-well plate (Greiner Bio-One #782900) and briefly centrifugated. The plates were sealed using a UV-transparent plate seal (Greiner, Merck #Z617571) to prevent evaporation and minimize exposure to atmospheric oxygen. Reactions were monitored by measuring the fluorescence polarization over time using the 485-520-520 fluorescence polarization filter (BMG Labtech) in a PHERAstar FS plate-reader instrument (BMG Labtech). Gain was auto-calibrated at each run to achieve ~25 % signal saturation in a control well, and was always found to be in the vicinity of 1400-1500, depending on the concentration of fluorescently labeled ubiquitin. Reactions were incubated for 6-12 hours at 30 °C. For each curve, the reaction rate was derived from the slope of the earliest curve region where the signal started rising monotonously, after the initial temperature equilibration. This was typically the case in the period 10-30 min after inserting the plate into the instrument.

### Single turnover RING activity assay

The E2 UbcH5 (UBE2D3) was charged in 500 μL volume by mixing 50 μM UbcH5, 1 μM Uba1, 60 μM ubiquitin, 6 μM Ubiquitin-DyLight488, and 5 mM ATP in 25 mM HEPES pH 7.5, 150 mM NaCl, 0.5 mM TCEP and 5 mM MgCl$_2$ buffer for 5 minutes at 37 °C. The conjugate was purified by size-exclusion chromatography using a Superdex 75 10/300 GL on an ÄKTA Pure FPLC system (Cytiva) into the same buffer and the concentration determined by the absorbance at 280 nm before use.

Mouse RNF213 RING domain (residues 3941-3999) was expressed with an N-terminal His-SUMO tag and C-terminal FLAG tag in E.coli and purified by HisTrap, ResourceQ ion exchange and size exclusion chromatography as described for the full-length RNF213 constructs. Single turnover discharge assays were carried out using 5 μM UbcH5 ~ Ub488/Ub, 5 μM RING, 20 mM lysine and were incubated at 37 °C and timepoints taken between 0-10 min by quenching with 1:1 with non-reducing SDS-PAGE loading dye. Controls in the absence of RING and the absence of lysine were included and treated as above.

### Preparation of the UBE2L3 ABP (Biotin-ABP)

Biotin-ABP was produced as described[32]. Briefly, Ub-thioester was generated from Ub(1–73)-intein, cleaved with 100 mM MESNA to give Ub-S-CH$_2$CH$_2$SO$_3$H, and purified by semi-preparative RP-HPLC (Mw 8417 Da). This was reacted with azideoethanamine to give Ub-azide (Mw 8361 Da). Ub-N$_3$ was conjugated with alkyne-functionalized TDAE by copper-catalyzed azide-alkyne cycloaddition, verified by LC−MS

(Mw 8624 Da). UBE2L3 (C17S/C137S) was mixed with Ub-TDAE and ABP assembly monitored by LC–MS and SDS-PAGE. Reactions were purified by size-exclusion chromatography.

### In vitro ABP labeling of RNF213

For quantitative ABP labeling measurements 768 nM RNF213 (ΔN338) and 25 µM Biotin-UBE2L3-ABP in 50 mM Tris-HCl pH 7.5, 2.5 mM $MgCl_2$, 150 mM NaCl, were incubated with 128 nM – 40 mM ATPγS (5-fold titration series), in the presence or absence of 900 µM ADP or AMP. All nucleotide stock solutions were prepared fresh in 50 mM Tris-HCl pH 8.0 and pH adjusted to pH 8.0. Reactions were incubated at 30 °C for 4 h, stopped by addition of LDS sample buffer, resolved by SDS-PAGE and analysed by Coomassie protein stain or fluorescent western blot.

### Adenylate charge

The adenylate charge[89] represents the relative concentrations of ATP and ADP, against the total adenine nucleotide pool (ATP, ADP and AMP):

$$\text{Adenylate charge} = (ATP + 1/2\,ADP)/(ATP + ADP + AMP)$$

In resting cells, the adenylate charge approximates to 1. Upon ATP consumption, ADP and AMP levels increase. Keeping the total adenine nucleotide pool at 1 mM, we incrementally decreased the adenylate charge from 1 to 0.7[90]. 1.5 µM RNF213 (ΔN338) was incubated with 68 µM UBE2L3 ABP in the presence or absence of varying concentrations of nucleotides ATPγS, ADP, or AMP, in 50 mM Tris-HCl pH 7.5, 2.5 mM $MgCl_2$, 150 mM NaCl. Reactions were incubated at 30 °C for 4 h before resolving on 3-8% Tris-Acetate gels.

### Tissue culture and cell line generation

**Maintenance.** 293 T and T-REx-293 cells were cultured (37 °C, 5% $CO_2$) in Dulbecco's modified Eagle's medium (DMEM) supplemented with 10% (v/v) fetal bovine serum (FBS), 2.0 mM L-glutamine and antibiotics (100 units/mL penicillin, 0.1 mg/mL streptomycin). THP-1 cells were cultured in Roswell Park Memorial Institute (RPMI) supplemented as above.

**Generation of T-REx-293[RNF213 KO].** To perform RNF213 activity measurements *in cellula*, an RNF213-null cell line was first generated. Oligonucleotides encoding an RNF213-specific guide RNA sequence as previously described[13] (GCTGAAAGCGGGCGCACTGC) were annealed and cloned into px459 (pSpCas9(BB)-2A-Puro (PX459) (a gift from Feng Zhang (Addgene plasmid # 48139; http://n2t.net/addgene:48139; RRID:Addgene_48139)). Sequence-verified plasmid was transfected into T-REx™-293 cells using polyethylenimine (PEI), transfectants were selected with 2.5 µg/mL puromycin 1 d later, then single-cell cloned by limiting dilution in 96 well plates 2 d after this, in the absence of puromycin. Twelve clones were screened by western blot using anti-RNF213 (HPA026790, Merck) and anti-β-Actin (Cell Signaling Technology, #3700). Genomic DNA was extracted from triaged clones (example of 3 clones in Supplementary Fig. 2a) using a DNeasy Blood & Tissue kit (Qiagen) and a PCR performed to amplify a region flanking the intended edit site (RH11, TCCTCTGAGGCAGCTGGTAT; RH12, ATGACAATCAGGCGGAAGTT). Amplified product was purified (Monarch PCR and DNA Cleanup Kit) and sequenced with nested primer RH9 (ATAACACGGGCACAGGATCT). Amplicons from the parental cell line were included as a control for the sequencing reaction. Clone 6 was chosen as the representative knockout clone and it contains a single adenosine insertion -3 from the target PAM site.

**Reconstitution of RNF213 variants in T-REx-293[RNF213 KO].** Full-length human RNF213 was cloned into pcDNA5™/FRT/TO (Thermo Fisher) and the full insert sequence verified. To achieve this RNF213 Isoform 3 was cloned by the sequential assembly of 3 fragments amplified by RT-PCR from human universal RNA. The first two fragments comprising nt 1 – 7087 and nt 7087 – nt11550 were amplified using primers adding a unique Nhe1 site upstream of nt1 and a unique Not1 site immediately downstream of nt11550, whilst also maintaining the native Sal1 site at nt7087. These two fragments were joined by conventional 3-way ligation (Nhe1-Sal1-Not1) using T4 DNA ligase into an intermediate vector. This product was then re-amplified using PCR primers designed to produce overlaps to a C-terminal third fragment (comprising nt11553-15595) amplified by RT-PCR and the components assembled by recombination using NEBuilder HiFi DNA Assembly kit (New England Biolabs Inc. Cat No. E5520S). Differences to the published NCBI sequence (NM_001256071.3) of the full-length final product included listed SNPs (at Q1133K (CAA > AAA), V1195M (GTG > ATG), D1331G (GAC > GGC), S2334N (AGT > AAT) as well as silent changes (at V1340 (GTC > GTG), A1550 (GCG > GCA and H4557 (CAC > CAT)). The full-length product was further re-amplified for incorporation into pcDNA5-FRT/TO vector by recombination as before. All RT-PCR products were produced using PrimeScript One step RT-PCR kits (Takara) and all PCR products using KOD DNA polymerase (Merck). All sequencing was performed by MRC PPU Sequencing Service, Dundee, Scotland.

The following primers were used.

| Step 1 | F amplification ATG | gtggctagcATGGAGTGTCCTTCGTGCCAGCATGTCTCCAAG |
|---|---|---|
| | R amplification native Sal1 site @7087 | gtgGTCGACATTGAAGGGCACCCTCTGGAGCAG |

| Step 2 | F amplification native Sal1 site @7087 | gtgGTCGACTTTGATAAACTGCCCAGACACAAG |
|---|---|---|
| | R amplification (end) no stop | gtggcggccgcTCTCATTTCTCGATTCCATTTCAGCACAGC |

| Step 3 | F amplification (overlaps 3 way product vector for NEB) | CGTTGGAACCATGAGCTGGCTGGATGTGAGATGACCCTG |
|---|---|---|
| | R amplification (overlaps 3-way product vector for NEB) | CTCGATTCCATTTCAGCACAGCAGCTGTTTTCCACACTG |

| Step 4 | F amplification 3-way product for NEB builder | tgctgaaatggaatcgagaaatgagagcggccgcttac |
|---|---|---|
| | R amplification 3-way product for NEB builder | CAGGGTCATCTCACATCCAGCCAGCTCATGGTTCCAACGGGC |

| Step 5 | F re-amplify from 65459 to overlap 41459 | ggtaccgagctcggatcccccggggaattcgctagcATGGAGTGTCCTTCGTGCCAGCATGTCTC |
|---|---|---|
| | R re-amplify from 65459 to overlap 41459 (with C-HA) | gccctctagactcgacttaggcgtaatctggcacatcgtatgggtaagcggccgcTCTCATTTCTCGATTCCATTTCAGC |

| Step 6 | F Mut repair 1 bp deletion at Gly2930 | GTCCAGGACCGAG TCCAAGGGTACTT TGCGTCCTTTGCCAA |
| | R Mut repair 1 bp deletion at Gly2930 | TTGGCAAAGGACGC AAAGTACCCTTGGAC TCGGTCCTGGAC |

Variants were generated by in vivo assembly (IVA) cloning[91]. The following primers were used.

| ΔRZ | AF1178 | TTCCAACCATGCCT GAAGACCAGACCGGC CACGTGCTGG |
| | AF1179 | GTCTTCAGGCATGG TTGGAAGAAAAGC |

| GAGA insertion | AF1201 | GGAGCAGGAGCACAG ACCGGCCACGTGCTGG |
| | AF1202 | GGTCTGTGCTCCTGC TCCGTCTTCAGGCAT GGTTGGAAGAAAAGC |

AAA3 + 4

| WA3 K2426A | AF1186 | AGAAACTGGCTGTGGGGCAACCAGG CTTATTAAATTCCTTAGCGACCTG |
| | AF1187 | TGCCCCACAGCCAGTTTCTCCC |
| WB3 E2488Q | AF1188 | GACACCATCTTGTT TTTTGATCAAGCCA ACACAACGGAAGCTATAAGC |
| | AF1189 | TTGATCAAAA AACAAGATGG TGTCCAACTGATG |
| WA4 K2775A | AF1190 | CCGGCAGCTCCGCG TCTCTCGCCAA GACCATCG |
| | AF1191 | CGCGGAGCTGCCGGGCTTC |

IVA PCRs were performed using 1 ng template, 100 nM primers, 200 μM dNTPs, 3% DMSO, 0.5 μL Herculase II Fusion DNA polymerase (Agilent). Cycle parameters were 95 °C 30 s, 18 cycles of 95°C 10 s, 60 °C 30 s, 72 °C 5 min. A final hold of 72 °C for 10 min was performed. 1 μL FastDigest DpnI (Thermo Fisher) was added and incubated on ice for 15 min. 1 μL reaction was added to 50 μL NEB® 5-alpha cells (NEB C2987H), incubated on ice 15 min, heat shocked at 42 °C 30 s, incubated on ice for 2 min, then 200 μL SOC media (NEB) added and cells recovered at 37 °C, 350 rpm, for 45 min. The complete mixture was plated on LB agar containing 100 μg/mL ampicillin. Colonies were screened by analytical restriction enzyme digest with HindIII and BamHI (Thermo Fisher) followed by sanger sequencing. 2.5 μg pcDNA5/FRT/TO-RNF213 was mixed with 2.5 μg pOG44, then added to 600 μL reduced serum media (Opti-MEM, Gibco) containing 25 μg linear, 25,000 MW polyethyleneimine (PEI) (Alfa Aesar) and incubated at room temperature for 20 min. DNA mixtures were added dropwise to 10 cm dishes of sub-confluent T-REx-293$^{RNF213\ KO}$ seeded the day before at 3×10$^6$ cells per dish. As a control, 5 μg pcDNA5/FRT/TO-RNF213 was transfected without pOG44 to control for non-specific integration. 2 d post transfection, cells were selected with 50 μg/mL Hygromycin B (Invivogen). When confluent, successful integration was confirmed by overnight 1 μg/mL tetracyclin treatment (Thermo Fisher

Scientific) followed by Western blot using anti-RNF213 (Merck HPA026790) and anti-β-Actin (Cell Signaling Technology #3700).

**Generation of Ube1L-depleted THP-1 cells.** To assess a role for ISG15 conjugation in RNF213 E3 activity, we first depleted UBA7/Ube1L from THP-1 cells. 3 sgRNA targets were generated using the CHOPCHOP server[92] – sg1 CGCACTAGGGCCTCATCCAG, sg2 CGCTTCGAAGC-TACTGGATG and sg3 TCTCTCGGTACTTGCGTGGT. Complementary oligos encoding these guide RNAs were annealed and cloned into plentiCRISPRv2 (a gift from Feng Zhang (Addgene plasmid # 52961; http://n2t.net/addgene:52961; RRID:Addgene_52961) and correct inserts confirmed by Sanger sequencing. HIV-1-based lentivectors were generated by co-transfection of HEK293T cells in 10 cm dishes with 5 μg HIV-1 GagPol expression plasmid (derived from HIV-1 molecular clone NL4-3), 5 μg plentiCRISPRv2-sgUbe1L expression vector (or an empty plentiCRISPRv2 plasmid that expresses Cas9 protein alone) and 1 μg VSV glycoprotein expression plasmid, using 55 μg poly-ethylenimine in 200 μL Opti-MEM. 24 h post transfection, 293 T media was refreshed. 48 h post transfection, supernatant was harvested, fil-tered (0.45 μm), aliquoted at stored at -70 °C. Non-adherent THP-1 cells were transduced with a titration of lentivirus-containing supernatant in the presence of 8 μg/mL Polybrene and incubated for 24 h, before centrifugation (5 min, 1000 xg), and resuspending in fresh RPMI media in the presence of 1 μg/mL puromycin. Selected cells were expanded and the dilution of lentivirus that gave ~ 50% viability under drug selection was selected for all further work, as this corresponds with ~ 1 Cas9/sgRNA cassette per genome. Ube1L depletion was con-firmed by Western blot (Ube1L polyclonal, 15818-1-AP, Proteintech) for each separate sgRNA, compared to the Cas9-only control cell line. While all three sgRNA produced successful Ube1L depletion, sgUbe-1L_3 was selected based on the most robust target depletion. Although not quantified, no growth differences were observed in any of the Ube1L-depleted cells lines. Control antibodies to measure the IFN transcriptional response were against OAS2 (Polyclonal, 19279-1-AP, Proteintech) or GAPDH (AM4300, Invitrogen). To assess ISG15 conjugation, membranes were probed for total ISG15 (Polyclonal, Cell Signaling Technology, 2743).

**Intracellular ATP measurements**
T-REx-293 or THP-1 cells were seeded at 2×104 cells per well in a 96-well plate. THP-1 cells were differentiated with 100 ng/mL PMA treatment overnight, followed by media change and further 24 h rest. Cells were stimulated with various interferon subtypes overnight or treated with 30 mM 2-DG for 3 h. ATP levels were measured using a commercial luciferase-based kit according to the manufacturer's protocol (Abcam, ab113849). Briefly, cells were washed once with PBS and subsequently 30 μL of supplied detergent was added to each well. The tissue culture plate was agitated on an orbital shaker for 5 min at 600-700 rpm. 30 μL of substrate was added to each well, and the plate was incubated for 5 min on the orbital shaker at 600-700 rpm. The plate was covered to allow dark adaptation, and luminescence was measured using a microplate luminometer (Glomax, Promega, and Hidex Chameleon). All experiments were performed with between 3-12 technical replicates at each condition.

**In cellula activity-based profiling of RNF213**
THP-1 cells were treated with 100 ng/mL phorbol-12-myristate-13-acetate (PMA) (Invivogen) overnight, the media was refreshed for a further 24 h, then cells were treated with 1000 U/mL IFNα (PBL-Bioscience) or mock-treated overnight. 3 hours before electropora-tion, THP-1 and T-REx-293$^{RNF213\ KO}$ expressing human RNF213 variants were treated with 10 μM MG132 (Merck) for 3 h, with or without 30 mM 2-deoxyglucose (Merck). Electroporations were performed using the Neon™ Transfection System, and 100 μL tips. 12 million THP-1 cells or 10 million T-REx-293 cells were used per condition. Cells were washed

in 1x PBS then resuspended in 150 µL Neon™ Resuspension Buffer R (THP-1) or Neon™ Resuspension Buffer T (T-REx-293), 24 µL Biotin-UBE2L3-ABP (prepared at 1 mg/mL), 40 µL 25 mM ATPγS pH 8.0 or buffer (50 mM Tris-HCl pH 8.0) and immediately electroporated twice at 1,400 V and pulse width 20 ms. Cells were directly transferred to pre-warmed media and incubated in humidified incubators at 37 °C, 5% CO$_2$ for 1 h, then pelleted at 300 $g$ and washed thrice in 1x PBS. Pellets were frozen at -20 °C until lysis. Cells lysates were extracted with ice-cold lysis buffer (50 mM Tris-HCl pH 7.5, 10 mM sodium 2-glycer-ophosphate, 50 mM sodium fluoride, 5.0 mM sodium pyrophosphate, 0.27 M sucrose, 50 mM NaCl, 0.2 mM phenylmethanesulfonyl fluoride (PMSF), 1.0 mM benzamidine, 10 µM TCEP, 1% NP-40) on ice for 20 min. Lysates were clarified by centrifugation at 4 °C for 10 min at 21,100 $g$. Supernatants were collected, protein concentrations determined by Bradford assay, and sample concentrations normalized. 10 µL equilibrated streptavidin magnetic beads (Pierce) were added to each sample and rotated for 2 h at 4 °C. Beads were washed twice in 0.2% SDS in 1x PBS, twice in 1x PBS, twice in 4 M Urea, twice in 1x PBS, then resuspended in 2x LDS sample buffer containing 2-mercap-toethanol, boiled at 90 °C 5 min, and loaded on pre-cast SDS-PAGE gels.

## Western blot

Samples were prepared in LDS sample buffer (Thermo Fisher Scientific) containing 2-mercaptoethanol. Electrophoresis was performed at 180 V in a NuPAGE 3-8% Tris-Acetate or 4-12% Bis-Tris gel (Invitrogen). Following electrophoresis proteins were transferred to PVDF membrane. Membranes were blocked in PBS-T (137 mM NaCl, 2.7 mM KCl, 10 mM Na$_2$HPO$_4$ 1.8 mM KH$_2$PO$_4$, 0.1% Tween-20) containing 5% (w/v) non-fat dried skimmed milk powder for 1 h, washed twice with PBS-T, and incubated with commercially available primary antibody in PBS-T containing 5% (w/v) bovine serum albumin (BSA, Fisher Scientific): for cell lysates, anti-RNF213 (Merck HPA026790) 200 ng, anti-Biotin-HRP (Cell Signaling Technology #7075) 1:5000 dilution, anti-UBA1 (Thermo Fisher PA5-81869) 150 ng; for recombinant protein experiments, anti-RNF213 (obtained by inoculating rabbits with peptide INELKVFVDLA-SISAGEND coupled to KLH carrier and purifying total immunoglobulin from blood serum, Eurogentec) 1:1000 dilution, Streptavidin Protein, DyLight™ 680 (Thermo Fisher 21848) 1:20,000 dilution. After washing in PBS-T, membranes were incubated for 1 h in secondary HRP-coupled antibody anti-Rabbit Cell Signaling Technology 7074S, 1:5000 dilution, anti-rabbit DyLight™ 800 4xPEG Conjugate (Cell Signaling Technology #5151) 1:10,000 dilution. Membranes were washed before adding chemiluminescent substrate (ECL Prime for T-REx-293 cells or ECL select for RNF213 in THP-1 cells, Cytiva Lifesciences) and exposed to radiographic film, or scanned using a LiCor Odyssey Scanner.

## Quantitative in cell activity-based proteomics

Peptides were obtained by initially electroporating biotin-ABP (1 mg/mL) with or without ATPγS (25 mM) into T-REx-293$^{RNF213\ KO}$ + RNF213 cells followed by streptavidin enrichment as described above. Loaded streptavidin resin was then treated with 1 µg sequencing-grade trypsin (Promega). Peptides were recovered on reverse-phase c18 spin columns (Thermo Fisher Scientific), eluted in 70% acetonitrile, and dried on a Savant SpeedVac (Thermo Scientific). Peptides were resuspended in 5% formic acid in water and 2µg were injected on a UltiMate 3000 RSLCnano System coupled to an Orbitrap Exploris 480 or an Orbitrap Fusion Lumos Tribrid Mass Spectrometer (Thermo Fisher Scientific). Peptides were loaded on an Acclaim Pepmap trap column (Thermo Fisher Scientific #164750) prior to analysis on a PepMap RSLC C18 analytical column (Thermo Fisher Scientific #ES903) and eluted on a 120 min linear gradient from 3 to 35% Buffer B (Buffer A: 0.1% formic acid in water, Buffer B: 0.08% formic acid in 80:20 acetonitrile:water (v:v)). Eluted peptides were then analysed by the mass spectrometer operating in data independent acquisition mode. Six biological replicates for each of the experimental conditions were analyzed sequentially. Peptides were searched against the Uniprot Swissprot Human database (released on 05/10/2021) using DiaNN (v1.8.1) operating in library free mode[93]. Statistical analysis was carried out using Python (v3.9.0) and packages pandas (v1.3.3), numpy (v1.19.0), sklearn (v1.0), scipy (v1.7.1), rpy2 (v3.4.5), Plotnine (v0.7.1) and Plotly (v5.8.2) and R (v4.1.3) and the package Limma (3.50.1)[94]. Protein groups identified with a less than 2 proteotypic peptides and quantified in less than 3 replicates were excluded. Missing values were then imputed using a gaussian distribution centered on the median with a downshift of 1.8 and a width of 0.3 (relative to the standard deviation) and protein intensities were median normalized. Protein regulation was assessed using two-tailed differential expression analysis carried out in LIMMA using an empirical Bayes-moderated linear model and P values were corrected using Benjamini-Hochberg multiple hypothesis correction. Proteins were considered significantly regulated if their corrected P-value was smaller than 0.05 and their fold change was greater than 1.5 or smaller than 1/1.5.

## Immunofluorescence microscopy

For the analysis of RNF213 variant expression, we chose Huh-7 cells for their abundance of lipid droplets. Cells were seeded on glass coverslips and transfected a day later with pcDNA5/FRT/TO-RNF213 variants. A day later, cells were fixed, permeabilised with PBS-Triton-X-100 (0.2%) and blocked with 5% bovine serum albumin in permeabilisation buffer. Coverslips were stained with anti-RNF213 (HPA026790) or Hoechst, before imaging on a Zeiss LSM 780 Confocal Microscope.

## Preparation of RNF213-ABP for XL-MS

A 600 µL reaction containing 1.5 µM full-length RNF213 (528 µg), 100 µM StrepII-UBE2L3 probe (1.56 mg) and 5 mM ATPγS were incubated in 50 mM Tris-HCl pH 7.5, 2.5 mM MgCl2, 150 mM NaCl at 30 °C for 4 h, with agitation (300 rpm). 1 µL of reaction before and after incubation were assessed by SDS-PAGE on a 3-8% Tris-Acetate gel.

The assembled RNF213-ABP adduct was further purified by SEC using a Superose 6 Increase 3.2/300 column (Cytivia) equilibrated with a buffer containing 200 mM KCl, 40 mM HEPES, 0.5 mM TCEP, pH 8.0. Fractions containing the adduct were identified by SDS-PAGE using a 4-20% Tris-Glycine gel (BioRad). This sample was also used for obtaining the cryo-EM structure of RNF213-ABP in absence of ATPγS.

## XL-MS analysis of RNF213-ABP

**Sample preparation.** 40 µL of the SEC-purified RNF213-ABP sample (0.3 mg/mL, buffer: 200 mM KCl, 25 mM HEPES, 2 mM TCEP, 10 mM MgCl2) was supplemented with urea to a final concentration of 8 M. Dithiothreitol (DTT) was added to 10 mM final concentration and the sample was incubated for 1 h at 37 °C. The alkylation was performed by adding iodoacetamide (IAA) to a final concentration of 20 mM and incubating for 30 minutes protected from light. The reaction was quenched by addition of DTT to 5 mM and incubated again 30 minutes at room temperature.

Then the sample was split in 2 equal parts:

The first part was diluted to 6 M urea with 100 mM ammonium bicarbonate (ABC) followed by addition of 500 ng Lys-C (FUJIFILM Wako Pure Chemical Corporation) and incubation at 37 °C for 2 hours. The sample was diluted to 2 M urea with 100 mM ABC, 500 ng trypsin (Promega, Trypsin Gold) were added and incubated at 37 °C overnight. The next day the sample was further diluted to 1 M urea and 750 ng Glu-C (Promega, V1651) were added and incubated for another 8 hours at 37 °C.

The second part was diluted to 1 M urea with 100 mM ABC, then 750 ng Glu-C (Promega, V1651) were added and incubated overnight at 37 °C. The next day 750 ng trypsin were added, and the sample was incubated for 8 h at 37 °C.

Both samples were acidified with trifluoroacetic acid (TFA, Pierce) to a final concentration of 1%. 20% of each sample was analyzed by LC-MS/MS.

**LC-MS/MS on Exploris 480.** Generated peptides were analyzed on an UltiMate 3000 RSLCnano system, which was coupled via a Nanospray Flex ion source to an Orbitrap Exploris 480 mass spectrometer equipped with a FAIMS pro interface (Thermo Fisher Scientific).

Peptides were loaded onto a trap column (PepMap C18, 5 mm × 300 µm ID, 5 µm particles, 100 Å pore size) at a flow rate of 25 µL/min using 0.1% TFA as mobile phase. After 10 min, the trap column was switched in line with the analytical column (PepMap C18, 500 mm × 75 µm ID, 2 µm, 100 Å, Thermo Fisher Scientific), which was operated at a flowrate of 230 nL/min at 30 °C. For separation a solvent gradient was applied, starting with 98% buffer A (water/formic acid, 99.9/0.1, v/v) and 2% buffer B (water/acetonitrile/formic acid, 19.92/80/0.08, v/v/v), followed by an increase to 35% B over the next 60 min, followed by a steep gradient to 90% B in 5 min, staying there for five min and decreasing to 2% B in another 5 min.

The Orbitrap Exploris 480 mass spectrometer was operated in data-dependent mode, performing a full scan (m/z range 375-1500, resolution 60,000, target value 3E6) at 3 different CVs (-45, -55, -65), followed by MS/MS scans to result in a cycle time of 1 second per CV. MS/MS spectra were acquired using a stepped normalized collision energy of 28 +/-2%, isolation width of 1.4 m/z, resolution of 30.000, maximum fill time of 100 ms, the target value of 2E5 and intensity threshold of 5E3 and fixed first mass of m/z = 120. Precursor ions selected for fragmentation (exclude charge state 1, 2, >8) were excluded for 30 s. The peptide match feature was set to preferred and the exclude isotopes feature was enabled.

**Data Analysis.** Raw files were analyzed using pLink 2.3.9 software. All MS/MS spectra were searched against a custom database containing only the 3 proteins Mys (based on mouse RNF213), ABP-UBE2L3 (based on human UBE2L3), and Ub-ABP (based on human ubiquitin). The peptide and fragment mass tolerances were set to ±10 ppm. The minimum peptide length was set to 6 amino acids, the maximum peptide mass to 10000 Dalton, the maximal number of missed cleavages was set to 3 and using combined trypsin and GluC as enzyme specificity (both without restriction at proline). ABP was defined as a cross-linker, with a mass of 306.18042 and a composition of H(22)C(14) N(6)O(2) on cysteine, serine, or threonine residues. The same mass was searched as a Monolink mass.

Carbamidomethylation on cysteine, oxidation on methionine, and deamidation on asparagine and glutamine were set as variable modifications. Data were filtered to 1% FDR by pLink and additionally to an E value < $10^{-4}$ on CSM level.

### Metal-binding assay

Synthetic peptides of the core RZ domain (murine residues 4446-4478) were solubilized in the RNF213 buffer (200 mM KCl, 25 mM HEPES, 0.25 mM TCEP, pH 7.2). Binding mixtures contained 200 µM of the zinc finger. For screening of different metals, 150 µM of each metal was used, or otherwise as indicated in the figure. For assaying metal binding of RZ mutants, 300 µM of $CoCl_2$ was used in all mixtures. Spectra were recorded for all mixtures using a small-volume 10 mm quartz cuvette (Hellma ultra Micro, Sigma Aldrich #Z627062) and a Denovix DS-11 FX+ Spectrophotometer, using no baseline correction, and using water as the blanking solution. The curves were baseline-corrected post hoc by setting the absorbance at 850 nm to zero.

### Cryo-EM data collection and analysis

The RNF213-ABP sample was recorded on a Titan Krios G3 instrument equipped with a Gatan BioQuantum Energy Filter and a Gatan K3 camera. Cryo-EM data processing was carried out in Relion 3.1, with particle picking carried out by a combination of Cryolo and Relion 3D template picking using the model generated by Cryolo-picked particles. 3D classification was additionally used to separate the particles representing ABP-coupled RNF213 ( ~ 2/3 of the particles) from free RNF213 ( ~ 1/3 of the particles). Images of the cryo-EM density were generated by ChimeraX.

### RNF213 model building

To generate the RNF213-ABP model, RNF213 (PDB: 6TAX) and UBE2L3 (PDB: 4Q5E) were docked into the unsharpened map using Chimera[95], and subjected to five cycles of real space refinement in Phenix (1.19.1)[96] using reference model restraints and morphing. The resulting model was then further refined using cycles of modeling and refinement with Coot (0.9.5)[97] and Phenix Real Space Refine.

### Generation of Alphafold models

The 3D molecular model of the RZ domain was generated by Alphafold2[62], using the RZ domain sequence. The model was manually adjusted to account for zinc coordination and to fit into the experimental density. RNF213 missing the N-terminal 340 residues was generated with Alphafold3.

### Analytical size-exclusion chromatography

To test for RNF213 oligomerization in the presence of ATPγS, 0.5 mg/mL RNF213 (500 µL) was incubated with or without ATPγS (2 mM) for 30 minutes. The samples were then resolved with a Superdex 200 10/300 GL column with SEC buffer (25 mM HEPES pH 7.2, 200 mM KCl, 0.25 mM TCEP) with an ÄKTA pure 25 instrument (Cytiva). Absorbance at 280 nm was recorded to plot an elution profile.

### Preparation of untagged CTD and GST-CTD domain

Residues 4986-5207 of human RNF213 were PCR amplified from full length human RNF213 cloned into the pcDNA5™/FRT/TO vector. The forward (5′-GCCTAGAATTCAACTATGAACATCTCTTTATGGACATC-3′) and reverse (5′-AATTTAGCGGCCGCTTATCTCATTTCTCGATTC-CATTTCAG-3′) primers were used, which introduced 5′ and 3′ EcoRI and NotI restriction sites, respectively. The PCR product and the vector pGEX-6P1 were double digested with EcoRI and NotI (Thermo Scientific) and ligated with T4 DNA ligase (NEB). The ligation reaction was transformed into DH5α cells and plasmid DNA from clones was isolated and a positive clone was confirmed by DNA sequencing. The plasmid product (pGEX-6P1-CTD) was transformed into BL21(DE3) cells (Novagen) and protein was expressed at 16 °C overnight following induction with 200 µM IPTG. Cells were harvested and lysed in buffer (20 mM HEPES pH 7.2, 200 mM KCl) and the clarified lysate was loaded onto glutathione sepharose 4B resin. The resin was washed with buffer (20 mM HEPES pH 7.2, 200 mM KCl, 1 mM DTT). Resin was left in 12 mL of buffer with 525 µg PreScission protease and incubated overnight at 4 °C. The supernatant was collected, concentrated with a centrifugal filter device (Amicon) and further purified in ITC buffer (20 mM $Na_2HPO_4$, 200 mM NaCl, 0.5 mM TCEP) by size exclusion chromatography using a HiLoad® 16/600 Superdex® 200 prep grade column connected to an ÄKTA Go FPLC system (Cytiva).

GST-CTD was purified as above except instead of PreScission protease treatment, the fusion protein was eluted with elution buffer containing 10 mM reduced glutathione). The fusion protein was then purified with a HiLoad® 26/600 Superdex® 75 pg column (Cytiva). Pooled fractions in buffer (50 mM $Na_2HPO_4$ pH 7.5, 150 mM NaCl 1 mM TCEP) were concentrated by centrifugal filtration and stored at -80 °C.

### GST-CTD pulldown assay

10 µM GST-CTD 'bait' was incubated with 10 µM UBE2D3 or UBE2D3 S22R C85K-Ubiquitin isopeptide for 15 minutes at 25 °C in buffer (40 mM Tris pH 7.5, 150 mM NaCl, 10 mM $MgCl_2$). 5 µL of pre-equilibrated GST-resin (10 µL slurry) was then added and the

contents were incubated at 25 °C for 1 hour with end-over-end inversion. Beads were pelleted, washed 3 times with buffer, before resuspension in 5 μL reducing LDS sample buffer and boiled at 95 °C for 5 mins. Input (IP), flow through (FT) and pulldown (PD) samples were then assessed by SDS-PAGE and Coomassie stain.

## Strep(II)-RNF213 pulldown assay

Full-length RNF213 with N-terminal Strep(II) tag (1 μM) was incubated with UBE2D3 or UBE2D3 S22R C85K-Ubiquitin isopeptide (10 μM) in buffer (40 mM Tris pH 7.5, 150 mM NaCl, 10 mM MgCl$_2$) supplemented with 5 mM ATP or ATPγS (Merck) as indicated for 15 mins 25 °C before adding 2.5 μL (5 μL slurry) Strep-Tactin Sepharose resin (IBA Life Sciences; 12723074) and incubating for a further 30 mins 25 °C with end-over-end inversion. Samples were washed 3 times with buffer supplemented with 1 mM ATP or ATPγS as appropriate, before incubation in 10 μL buffer containing 1U PreScission Protease for on-resin cleavage of the RNF213 Strep(II) tag for 1 hour at 37 °C (supplemented with 1 mM ATP or ATPγS where appropriate). Cleaved protein was then collected and assessed by SDS-PAGE and Coomassie staining, or anti-UBE2D3 immunoblotting (UBE2D1/2/3/4 antibody; ProteinTech; 68138-1-Ig; 1:5000).

## Isothermal Titration Calorimetry

UBE2L3 and UBE2L3 F63 were purified as described and purified by size-exclusion chromatography using ITC buffer and a HiLoad® 26/600 Superdex® 75 pg column (Cytiva). CTD domain and E2s were dialyzed for 48 h against ITC buffer at 4 °C. Proteins were concentrated with centrifugal filter devices (Amicon). The cell of a PEAQ-ITC calorimeter (Malvern Panalytical) was filled with UBE2L3 (70 μM) and the syringe was filled with CTD domain (715 μM). A titration was carried out consisting of 13 injections. No binding was detected. The Design Experiment tool within the Malvern software was used to simulate binding isotherms with the experimental concentrations, which suggested that a binding affinity of <500 μM should have generated a fittable curve, assuming a measurable enthalpy change was associated with binding.

## Statistical information

We used the following statistical tests. Details are provided in respective figure legends. In Fig. 1j, a non-linear regression was used to determine EC50 values and 95% confidence intervals; in Fig. 3e, a two-way ANOVA; in Fig. 3f, a one-way ANOVA; in Supplementary Fig. 1h, a two-way ANOVA; in Supplementary Fig. 2i, a two-tailed t test; in Supplementary Fig. 2j, a two-way ANOVA. Tests were performed in Graphpad Prism 10.

## Reporting summary

Further information on research design is available in the Nature Portfolio Reporting Summary linked to this article.

## Data availability

Atomic coordinates and cryo-EM density maps have been deposited in the Protein Data Bank (PDB) and EMDB databases under accession codes 7OIK EMD-12931 for the structure of the RNF213-ABP adduct, PDB 9I1J EMD-52571 for the structure of the RNF213-ABP with ATPγS, and PDB 9I1I EMD-52570 for the structure of the RNF213 WB34 mutant with ATP. The raw micrographs are available at the EMPIAR database (EMPIAR-10711,). The mass spectrometry proteomics data have been deposited to the ProteomeXchange Consortium via the PRIDE[98] partner repository with the dataset identifier PXD063291. Source data are provided with this paper.

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

## Acknowledgements

We thank all members of the Fletcher, Virdee and Clausen group for remarks on the manuscript and discussions. The cryo-EM data were collected at the cryo-EM platform of the European Molecular Biology Laboratory in Heidelberg, overseen by Felix Weis, and Diamond Light Source in Oxford. We thank Richard Imre from the Protein Chemistry Facility of IMP/IMBA for assistance with acquiring the data and analysis of the ABP cross-linking and Mathias Madalinski for preparing the synthetic peptides. We also thank Jana Neuhold for cloning the RNF213 expression constructs, and Thomas Heuser for general assistance with cryo-EM work, both from Vienna BioCenter Core Facilities. We thank Grahame Hardie of University of Dundee for valuable discussion and Guy Riddihough from Life Science Editors for help with editing. This work was supported by funding from a UKRI Future Leaders Fellowship (MR/T043482/1; A.F, A.B., R.H. and H.S.), the European Research Council (ERC) under the European Union's Horizon 2020 research and innovation programme (AdG 694978; T.C.), an FFG Headquarter Grant (No 852936; T.C.), the Austrian Science Fund (FWF, SFB F 79; V.F.). and the Vienna Science and Technology Fund (WWTF, GA No LS21-029 (L.E.B. and T.L.W.) and GA No LS21-009 (L.D.)). S.V., D.R.S. S.M. and N.T.W. received funding from the UK Medical Research Council (MC_UU_12016/8). S.V., A.F. and S.M. received funding from the Biotechnology and Biological Sciences Research Council (BB/P003982/1). S.V. and D.R.S. also received funding from a Wellcome Trust Discovery Award (225880/Z/22/Z). We also acknowledge pharmaceutical companies supporting the University of Dundee Division of Signal Transduction Therapy (Boehringer-Ingelheim, GlaxoSmithKline and Merck KGaA); and the Institute of Molecular Pathology (Boehringer Ingelheim).

## Author contributions

J.A., S.V., T.C. and A.F. designed the experiments, coordinated the project and wrote the manuscript. J.A. performed the biochemical, structural, and bioinformatic analyses. A.F. prepared ABP material, performed all biochemical experiments involving the ABP and prepared Ube1L-depleted THP-1 cells, with assistance from H.S.. A.F. and A.B developed and performed all *in cellula* experiments. A.B. and R.H. derived RNF213-negative and reconstituted cell lines. A.B. performed ATP measurements and confocal microscopy. D.B.G. built the molecular models with participation from T.C.. V.F., L.B., A.L., L.D., T.L.W. and A.M. purified RNF213 variants and performed biochemical experiments. E.R. performed the XL-MS analysis. D.R.S. carried out crystallization trials with the human RZ domain, pull down experiments and performed biochemical experiments. D.R.S. and S.M. prepared ABP material. F.L. analysed and processed quantitative DIA mass spectrometry data. S.V. performed ITC experiments. N.T.W. designed and cloned human RNF213 constructs.

## Competing interests

S.V. is an author of a patent relating to the ABP technology and is also Founder and shareholder of Outrun Therapeutics. The remaining authors declare no competing interests.
