## [Transparent Peer Review file · Nature Communications]

ATP functions as a pathogen-associated molecular pattern to activate the E3 ubiquitin ligase RNF213

Corresponding Author: Dr Adam Fletcher

Version 1:

Reviewer comments:

Reviewer #1

(Remarks to the Author)

The recent discovery of RNF213 as an E3 ligase that targets the LPS on the surface of cytosolic bacteria left many interesting questions to answer. Here Ahel and colleagues have found that RNF213 activity is controlled by cytosolic ATP levels, which are known to change in response to infection. Furthermore, they develop a new E3 activity assay, solve a new cryo-EM structure of RNF213 with an E2 enzyme and posit the existence of a new ATP-dependent family of E3 enzymes, which could respond to ATP as a damage or infection sensor. The paper is extremely elegant, the conclusions are well supported by the data and physiological relevance of the findings will appeal to a very broad audience.

I have several suggestions and questions to address to improve the manuscript.

1. The authors identify Cys4462 as a putative catalytic site - could they use a Cys4462Ser mutant to trap ubiquitin on the E3 in the structure?

2. The authors demonstrate that the Interferon mediated increase in ATP which activates RNF213 is reversible with 2DG. ISG15 modification (ISGylation) of proteins, which would also be induced under these conditions, has been shown to be present on oligomerized RNF213 on lipid droplets and to target enzymes in glycolysis which could also affect ATP levels in the cell. Can the authors map ISGylation of RNF213 under the conditions tested in Figure 3i via SDS-page to determine if blocking RNF213 activation with 2DG affects modification. Also could the authors block ISGylation through deletion or knock down of Ube1L to assess whether modification affects activity?

Where is the activated RNF213 localized? Does changing its activity relocalize it from lipid droplets to cytosol or change its oligomerization state?

3. Could the authors discuss their hypotheses of how active RNF213 could distinguish between docking on bacteria or lipid droplets? How do their findings relate to these distinct functions of the protein?

Reviewer #2

(Remarks to the Author)

The manuscript by Ahel et al. reported a novel regulatory mechanism of RNF213 ubiquitin ligase, namely through the binding of ATP. In addition to in vitro biochemical assays showing the stimulation of RNF213's activity, the authors also demonstrated the stimulation of RNF213's activity in live cells using an E3-based ABP introduced by electroporation. Furthermore, a cryo-EM structure of RNF213-E2 complex was obtained and revealed a new RZ domain in RNF213 that is responsible for transthiolation. Coupled with cross-linking mass spectrometry, they showed that Cys4462/His4483 residues locating on a loop in the RZ domain contribute to the catalytic activity of RNF213. Overall, this is an outstanding report that has addressed important questions in the regulation of E3 ubiquitin ligases in cellular defense against pathogens and linked this mechanism to intracellular ATP/ADP levels. This work also provides a new way of probing some E3 ligases in live cells. Below are a few questions aimed at further strengthening an already strong manuscript.

In Figure 3e, is the increase of WB3 mutant compared to WT statistically significant? If so, this indicates that the deficiency in hydrolysis of ATP due to the Walker B mutation E2488Q does not affect ATP binding. This mutant should be used in the

transthiolation reaction as shown in SI Fig. 1f to provide a convincing explanation of the lack of stimulation of WT RNF213 by ATP.

In Figure 4g and Supplemental Figure 1b, the lipid A-Ub product bands are very weak overall and not well resolved in some of the lanes. Is this level of E3 activity physiologically relevant?

In the cryo-EM structure of the RNF213-E2-Ub complex in the presence of ATP S, was the nucleotide resolved? The authors should include a figure/panel showing the nucleotide binding to the AAA module.

The authors identified three E3 ligases whose labeling by the Ub-E2 probe was upregulated by ATPγS. Do these E3s all contain an AAA ATPase module? TRIM28 was highlighted in the volcano plot in Figure 3k, presumably as an E3 ligase. It shows a high fold of enrichment and an acceptable P value. Why is it not accounted as a positive hit?

Figure 3c, the UBA1-ABP labeling band is missing in the middle WB panel.

Reviewer #3

(Remarks to the Author)

Reviewer #4

(Remarks to the Author)

In this manuscript, Ahel et al. focus on understanding catalytic and regulatory mechanisms of the E3 ubiquitin ligase, RNF213. The authors employed an innovative multidisciplinary approach to establish that the AAA3 and AAA4 ATPase sites of RNF213 serve as ATP sensors for the regulation of RNF213 ligase activity in vitro and in cells and that this feature is unique to RNF213 among E3 Ub ligases. The authors extend this finding to show that interferons stimulate the activity of RNF213 at least in part by increasing the concentration of ATP in the cell and providing a basis for involvement of this ligase in the innate immune response. Lastly, the authors obtained a cryo-EM reconstruction of an RNF213:E2-Ub ABP adduct which revealed a novel E2 binding module. While the RZ domain, which the authors posit harbors the catalytic cysteine residue of RNF213, is not visible in the cryo-EM reconstruction, residues important E2 binding and presumably transthiolation were identified. With that said, I was disappointed that the last third of the paper, which involves structural work, provided no insights of note into the mechanism of RNF213 regulation by ATP and instead pivoted to other aspects of RNF213 such as E2 binding and aspects of its catalytic mechanism which build from previous observations described in Ahel et al., *Elife* 2020. These issues are of course, important and interesting but they aren't fleshed out to the extent that ATP regulation is in the first 2/3 of the paper. Overall, this reviewer still finds the findings of the paper, in particular the regulatory mechanism involving ATP and identification of a new E2 binding module, to be novel and of significant interest to the field. I am supportive of publication in *Nature Communications* should the following important issues be adequately addressed:

1) What is the mechanism by which ATP regulates RNF213 activity? As noted above, the first 2/3 of the paper rigorously establishes that ATP regulates RNF213 in vitro and in cells. The next section involves cryo-EM studies and while a reader would expect some focus on the mechanism of RNF213 regulation there is no mention or focus on this issue at all. It was even difficult to ascertain the nucleotide state of RNF213 in the cryo-EM sample. Have the authors conducted cryo-EM analysis of RNF213 in the presence of ATP/Mg, AMP-PNP, ADP, and apo states etc... Are there nucleotide-dependent differences in the architecture of RNF213 that could provide a basis for how ATP stimulates its activity? Given that the authors have all of the reagents in hand, it is surprising that they would not have conducted such straightforward experiments that could provide insights into the regulatory mechanism. Lastly, the authors propose that ATP induces activating RNF213 oligomerization, but this should be visible by cryo-EM and testable using several other approaches.

2) The authors extend on findings from the Ahel et al., *Elife* 2020 paper, specifically the RING-independent activity of RNF213 and its tendency to function with UbCH7, an E2 involved in transthiolation. Here, the authors identified the E2 binding site and identified a putative catalytic cysteine and histidine residues. One important aspect of the proposed mechanism is the formation of a RNF213~Ub thioester intermediate. Despite the fact that these intermediates can be very difficult to isolate as evidenced by RBRs such as Parkin and ARIH1, the authors should note of what efforts were made to identify the RNF213~Ub intermediate, even in vitro. If there have not been any efforts to directly identify the intermediate, the relevant experiments should be conducted.

3) What is the role of the RING domain in the mechanism of RNF213 activity? Does the standalone RING domain harbor catalytic activity and if so, with what E2s? Is there a preference for lysine or cysteine? If there is no observable activity, is it possible that this RING functions with an E2 for another Ubl or is possibly involved in later steps of the process such as chain extension or branching? Following up on query 1, does the RING domain undergo nucleotide-dependent conformational changes when comparing this structure with that from Ahel et al. 2020?

4) Along those lines, does the RZ domain undergo nucleotide-dependent conformational changes? One can imagine a scenario where the RZ domain of apo or ADP bound RNF213 adopts an autoinhibitory conformation that is relieved upon ATP/AMP-PNP binding and accounts for its conformational variability in an active structure. Do the authors have any

insights into this?

5) It is a minor concern, but the E2 bound structure is covalently linked to RNF213. It is formally possible that this is an artifact of the crosslinking. Have the authors tested whether E2-Ub associates with RNF213 in manner consistent with the structure without crosslinking?

6) The manuscript has a section titled 'RNF213 is the only ATP-regulated E3 Ub ligase' yet in the discussion (lines 390/391) the authors speculate that ZNFX1 shares a similar ATP-dependent mechanism as RNF213. Please clarify.

Version 2:

Reviewer comments:

Reviewer #1

(Remarks to the Author)

Ahel et al have revised their manuscript entitled "ATP functions as a pathogen-associated molecular pattern to activate the E3 ubiquitin ligase RNF213." The authors use an activity-based probe to identify ATP-dependent enzymes using mass spectrometry. They turn their focus to the large antimicrobial E3 enzyme RNF213 and do a number of assays that show that its auto-ubiquitination and LPS modifying ability requires ATP. They solve a cryo-EM structure of the complex that reveals new information about E2 binding and activity.

The reviewers asked for a number of experiments concerning the structural effects of ATP binding, the localization of active RNF213, whether ISGylation controls oligomerization and whether activity correlates with docking on bacteria or lipid droplets.

The authors addressed the following experiments:

- Creating a Cys4462Ser (C4462S) mutant and evaluated its auto-ubiquitination and lipid transfer activity.
- Knocking out Ube1L and measured ATP levels in these cells and found that ATP production was lower in Ube1L-depleted cells compared to controls
- Showed ISGylation seemed to be critical for RNF213 ATP activity

Overall, the authors did a lot of experiments adding to the depth and breadth of the manuscript but the experiments on the localization of active RNF213 beg to be attempted with the tools the authors have developed. They say that the ABP does not provide localization but based on its biotin signal could the authors not use a fluorescently tagged streptavidin to localize active RNF213? What about PLA as an alternate approach (for RNF213 and ISG15)? They do not show their immunofluorescence staining to the reviewer as well, which would have been helpful, though in the manuscript the experiments are discussed but data is not shown. Furthermore, ISGylated RNF213 was detected in THP-1 cells by They and colleagues only after enriching lipid droplets. Could the authors enrich for lipid droplets in order to visualize ISGylated RNF213 following the method that They et al used? It seems like the authors hypothesize that local ATP concentrations generated by bacteria could activate RNF213 but right now the underlying data to support their findings are lacking. The data concerning ISGylation could instead support the hypothesis of ISG15 regulated complex formation so by not being able to localize activity it is hard to extrapolate the biological implications of the finding. Since the structures unfortunately did not trap bound ATP in part of RNF213 that was fully resolvable, nor did ATP binding provoke a large conformational shift in the structure it is hard to reconcile with the primary message of the paper. From an outside perspective it seems like the strength of the paper lies in the new technique to characterize E3 enzymes and less on the structural role that ATP binding plays.

Reviewer #2

(Remarks to the Author)

The authors have addressed the reviewer's points satisfactorily.

Reviewer #3

(Remarks to the Author)

Reviewer #4

(Remarks to the Author)

The authors have done an outstanding job addressing reviewer comments and have added considerable insights to what was already a very strong study. The novel ATP regulatory mechanism and E2 binding module of RNF213 are exciting and will be of great interest to the field. The authors' rigorous efforts are appreciated and this reviewer fully supports publication of the manuscript in Nature Communications.

Version 3:

Reviewer comments:

Reviewer #1

(Remarks to the Author)

I appreciate that the reviewers added the data previously not shown. While I thought the manuscript might be improved by localizing the activated RNF213 (via IF or biochemical methods), I appreciate the expanded explanation of why this is technically challenging. I am satisfied by the revised and expanded revision.

Reviewer 1

I have several suggestions and questions to address to improve the manuscript.

1. The authors identify Cys4462 as a putative catalytic site - could they use a Cys4462Ser mutant to trap ubiquitin on the E3 in the structure?

We have now cloned and purified a Cys4462Ser mutant of RNF213, aiming to trap a Ser~Ub conjugate. Below is a comparison of the auto-ubiquitination activities of WT and C4462S (CS), in the presence or absence of Lipid A substrate. We find that the CS mutant retains autoubiquitination activity to a substantial amount, whereas lipid transfer is undetectable. As RNF213 activity is abolished with an inert C4462A mutation (**Fig. 5e**), we conclude that serine can partially substitute for cysteine as an active site nucleophile. Such retention of activity is also observed with the downstream catalytic cysteine in the atypical E3 MYCBP2 (Pao et al. Nature 2018). The upstream cysteine that receives ubiquitin from E2, along with those in RBR E3s (as acknowledged by Reviewer 4), are also refractory to serine trapping. Moreover, due to the strong background of autoubiquitination activity observed with RNF213, it is impossible to discern a trapped ester adduct amongst the high molecular weight auto-modification adducts. We include the new data as **Supplementary Figure 9c**.

Supplementary Figure 9c) Autoubiquitination reactions for RNF213 Δ N300 and RNF213 C4462S. Although LPS ubiquitination is prevented with the C4462S mutation, it retains autoubiquitination activity. This background signal masks the potential formation of a stabilized ester-linked intermediate. Left, stainfree fluorescence (protein bands), and overlaid dylight fluorescence (labelled ubiquitin); center, dylight fluorescence; right Coomassie stain.

We have summarised our existing data below that supports C4462 being the active site nucleophile:

- RNF213 is functional with the E2 UBE2L3, that can only transfer ubiquitin to cysteine of transthiolation E3 ligases (Wenzel, Nature 2011)
- The labelling site of the E2~Ub ABP mapped to residue C4462 by crosslinking MS.
- A C4462A mutation fully ablates activity, whereas mutating its catalytic partner H4483A residues strongly reduces ubiquitination function.
- RNF213 Δ RZ is not labelled by the E2~Ub ABP in cells.

2. The authors demonstrate that the Interferon mediated increase in ATP which activates RNF213 is reversible with 2DG. ISG15 modification (ISGylation) of proteins, which would also be induced under these conditions, has been shown to be present on oligomerized RNF213 on lipid droplets and to target enzymes in glycolysis which could also affect ATP levels in the cell. Can the authors map ISGylation of RNF213 under the conditions tested in Figure 3i via SDS-page to determine if blocking RNF213 activation with 2DG affects modification. Also could the authors block ISGylation through deletion or knock down of Ube1L to assess whether modification affects activity?

Thanks for the suggestion. We have stably depleted Ube1L from THP-1 cells using CRISPR/Cas9. We screened 3 independent guide RNAs and selected one (sgUbe1L_3) which strongly depleted Ube1L (Figure R1). We present the characterisation data for sgUbe1L_3 in **Supplementary Figure 2e**, as this is the guide we take forward.

Figure. R1: Generation of Ube1L knockout cells (THP-1 macrophages) using sg1, sg2 and sg3 guide RNAs using CRISPR/Cas9 technology.

Supplementary Figure 2e) Stable depletion of Ube1L from THP-1 cells using CRISPR/Cas9. We screened 3 independent guide RNAs and selected one (sgUbe1L_3) which strongly depleted Ube1L. IFN stimulation of OAS2 was maintained in depleted cells.

For these experiments, we used bulk, puromycin-selected pools, rather than derive single cell clones – this was to avoid the potential for clonal variation in cellular ATP levels. We first characterised these cells by monitoring total ISG15 conjugation and ISG expression, upon IFN stimulation, in control cells expressing Cas9 alone, or the Ube1L-depleted cells expressing Cas9 and sgUbe1L_3. Confirming specific inhibition of the ISG15ylation cascade, we observed strong loss of ISG15 conjugates at all molecular weights, but not monomeric ISG15 (15 kDa) (**Supplementary Figure 2f**). IFN-induced expression of ISGs, evidenced by RNF213 and OAS2, was unaffected by Ube1L depletion, indicating an intact transcriptional response to IFN in these cells (**Supplementary Figure 2f**). We also treated cells with or without 30 mM 2-DG, as suggested, which in Figure 3 of the original paper we show reduces cellular ATP levels and reduces RNF213 ABP labelling. 2-DG treatment had no effect on total ISG15 conjugation, suggesting the effects of 2-DG are not via global impact on ISG15 conjugation (**Supplementary Figure 2f**). However, despite robust RNF213 enrichment by immunoprecipitation (not shown), we have been unable to detect ISG15-modified RNF213 in these cells by Western blot, so we have not been able to address whether ISG15 modification of RNF213 is affected by 2-DG treatment. Although we cannot rule out this possibility, we previously presented data (Supplementary Figure 2d) showing that the inhibition of RNF213 ABP labelling by 2-DG can be overcome by co-electroporating cells with ABP and ATPγS, 1 h before assessing RNF213-ABP labelling, suggesting that the defect in RNF213 activity caused by 2-DG is due to a transient decrease in ATP levels and is sensitive to nucleotide levels on short time scales.

Supplementary Figure 2f) ISG15 conjugation is qualitatively unimpaired by 2-DG treatment in WT THP-1 macrophages. IFN-dependent RNF213 upregulation is unaffected in Ube1L-depleted cells.

Using these two cell lines, we next measured basal cellular ATP levels and levels following IFN stimulation. As before, IFN stimulation raised intracellular ATP levels significantly, and this occurred in control or Ube1L-depleted cells (**Supplementary Figure 2i,j**). However, in Ube1L-depleted cells, cellular ATP levels were significantly lower than in control cells and IFN-induced ATP levels did not rise significantly higher than untreated control cells.

To address whether ISG15 modification (of anything) affects RNF213 activity, we performed our in-cell E3 activity assay in control or Ube1L-depleted cells by electroporating Biotin-ABP into the above cells. As shown in Figure 3 of the current paper, IFN treatment causes a robust activation of the RNF213 E3, above and beyond the induction of RNF213 protein expression (**Supplementary Figure 2g**). However, post-translational activation of E3 activity was substantially reduced in cells lacking ISG15 conjugation ((**Supplementary Figure 2g**). Potentially, this is due to the lower ATP levels achieved upon IFN stimulation of Ube1L-depleted cells (**Supplementary Figure 2i,j**); ATP levels here do not significantly exceed the levels measured in control, untreated cells. Alternatively, this reflects a requirement for ISG15 modification (target unknown) in IFN-induced RNF213 E3 activation. The latter model would support previous reports from They et al. (Nat Comms, 2021), that ISG15 conjugation is required for RNF213 oligomerisation and activation/recruitment to lipid droplets. We attempted to visualise RNF213 localisation to lipid droplets in differentiated THP-1

macrophages upon IFN treatment, by immunofluorescence microscopy, however, we were unable to detect this; rather RNF213 was diffusely expressed throughout the cell (data not shown).

Supplementary Figure 2i) Reduced ATP levels are found in THP-1 cells upon Ube1L depletion. Two-tailed t test, $P < 0.0001$; error bars are SD ($n = 12$). **j)** Quantification of ATP levels in WT and Ube1L-depleted macrophages upon IFN treatment, representative data of $n = 3$. Two-way ANOVA, P values compare all conditions to untreated control THP-1 cells (left to right); 0.0075 (**), 0.6003 (ns), 0.0175 (*), 0.5895 (ns), 0.2334 (ns), 0.9879 (ns); error bars are SD ($n = 6$).

Supplementary Figure 2g) THP-1 cells treated with or without IFN-I overnight, and electroporated with a buffer control or biotin-ABP. Streptavidin enriched material probed with anti-RNF213 antibodies, indicative of RNF213 E3 activity. A reduced IFN-dependent increase in ABP signal for RNF213 is indicative of posttranslational activation of RNF213 being impaired in Ube1L-depleted cells.

Where is the activated RNF213 localized? Does changing its activity relocalize it from lipid droplets to cytosol or change its oligomerization state?

We don't yet have a good understanding of where activated RNF213 is localised, and we have been unable to visualise recruitment of endogenous RNF213 to lipid droplets in these differentiated THP-1 cells. Please note that in our ABP assays, we only obtain information on the bulk activation status of RNF213 in a cell lysate (containing both activated and latent RNF213), while spatial information is lost. This would require specific fractionation approaches and/or design of ABPs that demonstrate fluorescence upon RNF213 labelling, which is a challenging project by itself, beyond the scope of this work. The same holds true for connecting activity status and oligomerisation behaviour in cells, which is complicated by the fact that IFN itself elevates ATP levels and thus activation of RNF213, while in parallel inducing formation of higher order RNF213 forms. Thus, it is extremely challenging to distinguish between correlation and causality.

3. Could the authors discuss their hypotheses of how active RNF213 could distinguish between docking on bacteria or lipid droplets? How do their findings relate to these distinct functions of the protein?

Rather than binding to certain cellular structures, we propose RNF213 senses subcellular ATP/ADP ratios to become active in discrete environments, like in proximity of cytosolic bacteria. We are inspired by the work from Albert Pol describing lipid droplets as central defence hubs that coordinate antimicrobial programmes by gathering stimulation-specific proteomes (Bosch et al., Science, 2021). In their model, lipid droplets act as vehicles to deliver immune proteins to their microbial targets in the cell. Our data suggest that fluctuations in ATP/ADP ratios occur during infections, and we hypothesise that the surface of lipid droplets, or cytosolic bacteria, experience such local ATP/ADP fluctuations, which adjusts the E3 activity of proximal RNF213 molecules. Our data do, so far, not inform on downstream consequences of RNF213 E3 activation; possibly, the ubiquitination of diverse targets at each location (e.g. LPS ubiquitination vs ubiquitination of lipid droplet target) related to distinct downstream signaling events (xenophagy vs lipophagy). Conversely, it is also possible that RNF213 ubiquitinates a conserved target at each site, and the downstream events are

similar. We still lack sufficient understanding of the specific targets of RNF213 and how their ubiquitination inhibits pathogen replication, as well as the role of RNF213 in relation to lipid droplets. For all these reasons, we decided to focus our Discussion on the new targeting mechanism (sensing ATP as PAMP), rather than speculating on possible restriction mechanisms.

Reviewer #2

The manuscript by Ahel et al. reported a novel regulatory mechanism of RNF213 ubiquitin ligase, namely through the binding of ATP. In addition to in vitro biochemical assays showing the stimulation of RNF213's activity, the authors also demonstrated the stimulation of RNF213's activity in live cells using an E3-based ABP introduced by electroporation. Furthermore, a cryo-EM structure of RNF213-E2 complex was obtained and revealed a new RZ domain in RNF213 that is responsible for transthiolation. Coupled with cross-linking mass spectrometry, they showed that Cys4462/His4483 residues locating on a loop in the RZ domain contribute to the catalytic activity of RNF213. Overall, this is an outstanding report that has addressed important questions in the regulation of E3 ubiquitin ligases in cellular defense against pathogens and linked this mechanism to intracellular ATP/ADP levels. This work also provides a new way of probing some E3 ligases in live cells. Below are a few questions aimed at further strengthening an already strong manuscript.

In Figure 3e, is the increase of WB3 mutant compared to WT statistically significant? If so, this indicates that the deficiency in hydrolysis of ATP due to the Walker B mutation E2488Q does not affect ATP binding. This mutant should be used in the transthiolation reaction as shown in SI Fig. 1f to provide a convincing explanation of the lack of stimulation of WT RNF213 by ATP.

Thanks for the comment. Yes, the differences in ABP labelling between all RNF213 variants and the WT are statistically significant; apologies for not including a statistical test in the original paper. We have updated **Figure 3e** to include the two-way ANOVA analysis.

Figure 3e) Quantification (densitometry) of fold increase in ABP labelling comparing ATPγS- to buffer-electroporated control, from **d** ($n = 3$). Two-way ANOVA, P values compare the variants to WT (left to right); 0.0371 (*), 0.0391 (*), 0.0133 (*), 0.0038 (**); error bars are SD.

We agree with the reviewer in their interpretation that the deficiency in ATP hydrolysis as seen in the single mutants WB3, WB4 and, in particular, the double mutant WB34 means that ATP (or ATPγS) binding is intact in these variants, supporting our hypothesis that ATP binding, but not hydrolysis, is necessary for E3 activation (Fig. 1 and Supplementary Figure 1). To test our prior assumption (page 6, Supplementary Figure 1f) that ATP was not able to activate RNF213-ABP labelling compared to ATPγS, presumably due to a narrower RNF213 activation window with the rate of ATP hydrolysis being incompatible with the slow kinetics of RNF213-ABP labelling, we have performed the following experiments (**Supplementary Figure 1g,h**): We incubated RNF213 WT or WB4 proteins with our UBE2L3-based ABP, *in vitro*, in the presence of increasing concentrations of ATP. We also increased the highest nucleotide concentration to 40 mM (10 mM previously) (Supplementary Figure 1f). Quantitative fluorescent Western blots of these reactions revealed that 1) Activation of WT RNF213 can be measured with our ABP at elevated ATP concentrations, and 2) WB4 protein is ~100-fold more sensitive to ATP than WT protein, supporting the hypothesis that reducing ATP hydrolysis allows sustained RNF213 activation by ATP, and explaining the general observation that ATPγS was a more potent activator of E3 activity than ATP. We would like to note that the difference between ATPγS and ATP on E3 activity is most apparent in the ABP labelling assay, where kinetics are more influential on outcome; in E2~Ub discharge assays that involve native transthiolation, which is rapid,

ATP is still less effective than ATP γ S in activating RNF213, but only by ~ 10-fold (Figure 1d).

Supplementary Figure 1g) Recombinant RNF213 and RNF213 WB4 were tested for transthiolation activity with a biotin-tagged ABP *in vitro* in the presence of increasing concentrations of ATP. * Each data point was performed in duplicate. The WB4 mutant is labelled with the probe more efficiently at low and intermediate ATP concentrations than WT. As the ABP activity signal is higher at elevated ATP concentrations, sustained activation is likely due to this displacing the inhibitory ADP product, further underscoring the importance of the effect of ATP/ADP ratio on RNF213 transthiolation activity. **h)** Quantification of ABP signals obtained in **g**. Two-way ANOVA, P values (left to right); 0.683 (ns), 0.0002 (***), 0.0327 (*), 0.0565 (ns); significance threshold 0.05.

In Figure 4g and Supplemental Figure 1b, the lipid A-Ub product bands are very weak overall and not well resolved in some of the lanes. Is this level of E3 activity physiologically relevant?

We agree with the reviewer, the abundance of lipid A-Ub adduct is very weak compared to RNF213 autoubiquitination. While it is difficult to compare this *in vitro* system with what might occur upon the bacterial membrane in a cell, where additional factors may enhance recruitment to LPS and increase ubiquitination efficiency, we also note that the full details of LPS ubiquitination have not been fully elucidated, and the stoichiometry of lipid A modification vs modification of alternative targets, including RNF213 itself, have yet to be understood *in vivo*. Nonetheless, we show that ATP/ADP ratios control the upstream transthiolation activity, which would affect all RNF213 substrates. Indeed, the appearance of the Lipid A-Ub adduct that we observe correlates well with the relative *in vitro* and *in cell* E3 activities across the panel of

RNF213 variants, so we believe the modification is a faithful reflection of RNF213 target modification.

In the cryo-EM structure of the RNF213-E2-Ub complex in the presence of ATP γ S, was the nucleotide resolved? The authors should include a figure/panel showing the nucleotide binding to the AAA module.

The sample of the ABP complex was purified by size-exclusion chromatography before making EM grids, which removed unbound ATP γ S. We presumed ATP γ S would remain in the active sites of AAA3 and AAA4, as the activated state – with nucleotides bound – should have been preserved once the ABP probe got covalently linked to RNF213. However, this assumption turned out to be false and no nucleotide was present in AAA3 or AAA4. We therefore determined a further cryoEM structure of RNF213 in complex with E2-Ub ABP. Rather than purifying the RNF213-ABP complex, this time we applied the ATP γ S-stimulated reaction directly to EM grids (ATP γ S *in situ*). In the resulting 4 angstrom cryoEM structure, we observed nucleotides in AAA3 and AAA4, however in both cases the extra density was not well defined. Though we incubated the sample with ATP γ S, the density in both active sites matched best to ADP nucleotides, likely presenting the hydrolysis products accumulating during the 3-4 hours of EM sample preparation. We include these new structural data as **Supplementary Figures 6, 7 and Table 1**. In the discussion of this structure (see below), we included a corresponding panel that illustrates the nucleotide binding state of the separate ATPase sites.

Supplementary Figure 6

Supplementary Figure 6. Cryo-EM analysis of RNF213-ABP in complex with ADP. **a)** Example micrograph, low-pass filtered for visual clarity. **b)** Image class averages as generated by 2D classification in Relion. **c)** Angular distribution heat map of particles used to reconstruct the cryo-EM density. **d)** Fourier Shell Correlation (FSC) curves indicating a resolution of 3.8 Å using the FSC=0.143 criterion. **e)** Reconstructed cryo-EM density colored by local resolution. The resolution color scale is indicated. **f)** Angular distribution heat map of particles used to calculate a focused cryo-EM density map of the AAA core. **g)** Fourier Shell Correlation (FSC) curves indicating a resolution of 3.4 Å using the FSC=0.143 criterion. **h)** Reconstructed cryo-EM density colored by local resolution. The resolution color scale is indicated.

Supplementary Figure 7

Supplementary Figure 7. Structural alignment of CryoEM RNF213-ABP complex with second RNF213-ABP CryoEM model obtained with ATP γ S *in situ*. **a)** The structure obtained from applying the unpurified ABP reaction (ATP γ S *in situ*) to cryo-EM grids underwent modest structural rearrangement (RMSD = 2.66 Å) relative to the original structure. Alignments were carried out with PyMOL 3.0.0 (Schrödinger). **b)** As consistently observed before, density corresponding to ATP was present at the AAA2 site that lacks Walker B residues required for hydrolysis. The *in situ* ATP γ S conditions produced density consistent with ADP being bound to sites AAA3 and AAA4.

The authors identified three E3 ligases whose labeling by the Ub-E2 probe was upregulated by ATP γ S. Do these E3s all contain an AAA ATPase module? TRIM28

was highlighted in the volcano plot in Figure 3k, presumably as an E3 ligase. It shows a high fold of enrichment and an acceptable P value. Why is it not accounted as a positive hit?

Thanks for this comment – no, in fact these three additional E3s (ITCH, UBE3A and RNF19A) do not contain AAA modules. We interpret their increased E3 activity as an indirect consequence of their partnership with AAA proteins. Firstly, UBE3A and RNF19A both interact with the 19S proteasome which comprises a ring of AAA ATPases – the 19S activity is plausibly attenuated by ATP γ S, and might be coupled to associated E3s (e.g. Buel *et al.*, *Nat Comm*, 2020). Secondly, ITCH is shown to interact with and ubiquitinate the AAA ATPase Vps4A (Deng, *Journal of Virology*, 2022) during viral infection, which stimulates Vps4A ATPase activity, suggesting a possible regulatory link. Detection of TRIM28, despite showing a high fold enrichment, is at the significance threshold, so we did not consider this as a true hit. TRIM28 is also not known to harbour a catalytic cysteine, so we hypothesise that this is a non-specific enrichment.

Figure 3c, the UBA1-ABP labeling band is missing in the middle WB panel.

Yes, in this experiment Uba1~ABP band was weak and only revealed by high exposure. Here is a revised **Figure 3c** to include a long exposure image revealing the Uba1~ABP band.

Figure 3e. Revised figure including a high exposure blot showing weak Uba1-ABP signal.

Reviewer #3

Reviewer #4

In this manuscript, Ahel et al. focus on understanding catalytic and regulatory mechanisms of the E3 ubiquitin ligase, RNF213. The authors employed an innovative multidisciplinary approach to establish that the AAA3 and AAA4 ATPase sites of RNF213 serve as ATP sensors for the regulation of RNF213 ligase activity in vitro and in cells and that this feature is unique to RNF213 among E3 Ub ligases. The authors extend this finding to show that interferons stimulate the activity of RNF213 at least in part by increasing the concentration of ATP in the cell and providing a basis for involvement of this ligase in the innate immune response. Lastly, the authors obtained a cryo-EM reconstruction of an RNF213:E2-Ub ABP adduct which revealed a novel E2 binding module. While the RZ domain, which the authors posit harbors the catalytic cysteine residue of RNF213, is not visible in the cryo-EM reconstruction, residues important E2 binding and presumably transthiolation were identified. With that said, I was disappointed that the last third of the paper, which involves structural work, provided no insights of note into the mechanism of RNF213 regulation by ATP and instead pivoted to other aspects of RNF213 such as E2 binding and aspects of its catalytic mechanism which build from previous observations described in Ahel et al., *Elife* 2020. These issues are of course, important and interesting but they aren't fleshed out to the extent that ATP regulation is in the first 2/3 of the paper. Overall, this reviewer still finds the findings of the paper, in particular the regulatory mechanism involving ATP and identification of a new E2 binding module, to be novel and of significant interest to the field. I am supportive of publication in *Nature Communications* should the following important issues be adequately addressed:

1) What is the mechanism by which ATP regulates RNF213 activity? As noted above, the first 2/3 of the paper rigorously establishes that ATP regulates RNF213 in vitro and in cells. The next section involves cryo-EM studies and while a reader would expect some focus on the mechanism of RNF213 regulation there is no mention or focus on this issue at all. It was even difficult to ascertain the nucleotide state of RNF213 in the cryo-EM sample. Have the authors conducted cryo-EM analysis of RNF213 in the presence of ATP/Mg, AMP-PNP, ADP, and apo states etc... Are there nucleotide-dependent differences in the architecture of RNF213 that could provide a basis for how ATP stimulates its activity? Given that the authors have all of the reagents in hand, it is surprising that they would not have conducted such straightforward experiments that could provide insights into the regulatory mechanism. Lastly, the authors propose that ATP induces activating RNF213 oligomerization, but this should be visible by cryo-EM and testable using several other approaches.

We agree with the referee that it would have been ideal to include a detailed mechanism of ATP-dependent regulation. Unfortunately, resolving the precise

mechanism has proven to be extremely challenging, even with all reagents at hand. We have generated two new structures using conditions that might trap the fully activated state. Firstly, we applied the ABP reaction directly to the EM grids, thereby retaining excess ATP γ S. This yielded a comparable structure, albeit with new density allowing the modelling of ADP in sites AAA4 and AAA4 (please refer to Reviewer 2 section). Disappointingly, there were no apparent changes consistent with elevated transthiolating activity. Secondly, we solved a structure of the isolated RNF213 WB34 (hydrolysis defective) in the presence of ATP. With this structure we could model ATP in site AAA4 and AAA3 was empty. We include these new data as **Supplementary Figures 8, 9 and Table 1**. In this structure there are also no apparent structural changes consistent with elevated transthiolation activity. Unfortunately, in both of our new structures the RZ domain could not be modelled. These challenges are reminiscent of the molecular switch in the AAA core of dynein, where decades of research have not fully resolved its allosteric behavior. Our extensive efforts reveal that a concise answer to the referee's question is beyond the aim and time scale of the present study.

We can only conclude that this an extremely challenging intermediate to resolve in its entirety, which will be key for complete elucidation of the transfer mechanism. We have added to the discussion that activation may require modest structural changes we have been unable to resolve, or the process is highly transient.

Regarding ATP-dependent changes in oligomeric states, the logical scenario is ATP binding, previously shown to induce cellular RNF213 self-association, triggers transthiolation activity. However, our additional cryoEM grids (ATP γ S *in situ* RNF213-E2-Ub and AAA mutants) present no evidence of oligomerisation and neither does size-exclusion chromatography (**Supplementary Figure 2h**). We add the following to the discussion relating to this point:

Intriguingly, mirroring its requirement for the activation of transthiolation, ATP binding induces RNF213 self-association in cells³⁸. Although activation of transthiolation through oligomerization is a compelling hypothesis, we could neither observe ATP-dependent oligomerisation structurally nor biochemically, hence this may be reserved for membrane targeting⁸⁷.

Supplementary Figure 2h) Size-exclusion chromatography of RNF213 alone or RNF213 pretreated with ATP γ S (2 mM). Separation was carried out with a Superdex 200 10/300 GL column (Cytiva).

In summary, while we acknowledge that a detailed mechanism of ATP-dependent regulation would be valuable, resolving this is a long-term challenge. We believe the data presented provide important foundational insights and highlight the complexities of ATP regulation in RNF213.

2) The authors extend on findings from the Ahel et al., Elife 2020 paper, specifically the RING-independent activity of RNF213 and its tendency to function with Ubch7, an E2 involved in transthiolation. Here, the authors identified the E2 binding site and identified a putative catalytic cysteine and histidine residues. One important aspect of the proposed mechanism is the formation of a RNF213~Ub thioester intermediate. Despite the fact that these intermediates can be very difficult to isolate as evidenced by RBRs such as Parkin and ARIH1, the authors should note of what efforts were made to identify the RNF213~Ub intermediate, even in vitro. If there have not been any efforts to directly identify the intermediate, the relevant experiments should be conducted.

As discussed for Reviewer 1, we have now cloned and purified a Cys4462Ser mutant of RNF213 and have attempted to trap a Ser~Ub conjugate. As shown above, we compared the ubiquitination activities of WT and C4462S (CS) and observed that the C4462S mutant retains E3 autoubiquitination activity (lipid activity was undetectable). Retention of catalytic activity with a serine mutant is preceded when there is a proximal general histidine base that can deprotonate the foreign OH group. The rapid formation of high molecular weight autoubiquitination products obscures the detection of a potential Ub adduct at Ser4462. Overall, these data highlight the dynamic nature

of the ubiquitin transfer mechanism in RNF213, complicating the trapping and isolation of the reaction intermediate. A brief discussion of these findings and the inherent challenges has been added to the manuscript.

3) What is the role of the RING domain in the mechanism of RNF213 activity? Does the standalone RING domain harbor catalytic activity and if so, with what E2s? Is there a preference for lysine or cysteine? If there is no observable activity, is it possible that this RING functions with an E2 for another Ubl or is possibly involved in later steps of the process such as chain extension or branching? Following up on query 1, does the RING domain undergo nucleotide-dependent conformational changes when comparing this structure with that from Ahel et. al. 2020?

We do not yet understand what role the RING plays in the E3 mechanism of RNF213. We previously found that RNF213 lacking the RING domain had qualitatively similar activity to full-length, with both UBE2L3 and UBE2D3. To further address this question, we have cloned and purified the RING domain of RNF213 and found it is unable to act as an E3 in isolation. As observed in E2~Ub (UBE2D3) discharge assays, in the presence or absence of high concentrations of lysine; the discharge of Ub from UBE2D3~Ub is not facilitated by the RNF213 RING (**Supplementary Figure 5a**). Consistent with these data, the RNF213 RING lacks a classic linchpin residue and contains a hydrophobic leucine instead. Functional linchpins activate the E2~Ub by stabilising the reactive closed conformation by forming hydrogen bonds, typically via a basic residue such as Arg or Lys found immediately after the last zinc-coordinating cysteine residue. Hence, this data confirms that the RNF213 RING does not behave as a typical RING domain. In cells, we cannot formally exclude the possibility that E2 binding to the RING regulates the CTD-RZ activity we characterise herein.

With regards to induced conformational changes upon ATP binding to the AAA core – of note, the AAA module and RING domain are located on the exact opposite sides of the RNF213 macromolecule - we did not observe such rearrangement.

Supplementary Figure 5a) The ability of the RNF213 RING domain to stimulate discharge of Ub from an E2~Ub to free lysine (20 mM), a characteristic of archetypical RING domains, was tested. Reaction was performed under single turnover conditions (UbcH5~Ub (5 μ M) and His-SUMO-RNF213_{RING} (5 μ M)). Left, signal from DyLight488 (Ub); right, Coomassie stain

4) Along those lines, does the RZ domain undergo nucleotide-dependent conformational changes? One can imagine a scenario where the RZ domain of apo or ADP bound RNF213 adopts an autoinhibitory conformation that is relieved upon ATP/AMP-PNP binding and accounts for its conformational variability in an active structure. Do the authors have any insights into this?

We have not been able to structurally characterize the RZ domain in our cryo-EM complexes. Similarly, crystallography attempts using isolated mouse and human RZ domains across hundreds of conditions did not yield crystals. This suggests that the RZ domain may be conformationally dynamic, consistent with a possible role in nucleotide-dependent regulation. In our cryo-EM data, weak and poorly defined density was observed in the expected region of the RZ domain, but the quality of this density was insufficient to incorporate the catalytic domain. Based on the AF3 model, the RZ domain is positioned such that its catalytic residues (Cys4462 and His4483) are oriented towards the protein interior rather than the E2~Ub conjugate in the ABP

complex. This implies that the RZ domain may detach or reorient from the protein body to engage in the transthiolation reaction. To account for this prediction, we have included **Supplementary Figure 11b** showing the AF3 model with the RNF213-E2 complex, illustrating the spatial arrangement of the RZ domain. This supports the likelihood of a reorientation being necessary for ubiquitin transfer. While we cannot conclude if nucleotide binding directly regulates these changes, our findings are consistent with the RZ domain being a flexible and dynamic component of RNF213.

5) It is a minor concern, but the E2 bound structure is covalently linked to RNF213. It is formally possible that this is an artifact of the crosslinking. Have the authors tested whether E2-Ub associates with RNF213 in manner consistent with the structure without crosslinking?

We appreciate the concern. As we could not prepare sufficient full-length RNF213 for biophysical affinity measurements, we have purified GST-tagged CTD (human RNF213 residues 4986-4207) and used this in pulldown experiments with either WT or F63A mutant human UBE2L3, however we detected no binding to E2 (**Supplementary Figure 4b**). We also attempted isothermal titration calorimetric measurement of the interaction between the untagged human CTD domain with UBE2L3 (**Supplementary Figure 4c,d**). We detected no binding. Assuming the binding event is associated with a measurable enthalpy change, simulated isotherm generation using the experimental cell and syringe concentrations suggested the affinity must be $> \sim 500 \mu\text{M}$. Speculating that affinity might be driven by additional interactions beyond the CTD domain and E2 (e.g. E3 shell, RZ domain, ubiquitin), we carried out pull downs with full length RNF213 against UBE2D3 and UBE2D3~Ub* (* a stable isopeptide linked C86K mutant carrying a backside binding S22R mutation). Pull downs were also performed in the presence of ATP and ATP γ S (**Supplementary Figure 4e,f**). We could not detect pull down of E2 or E2~Ub* under any of the conditions. Nevertheless, to establish if additional contacts might be required for catalytic activity, we have purified a new shell mutant (R4617A) proximal to the secondary E2 interacting site that might destabilise this region and detected a loss of RNF213-autoubiquitination (**Figure 3d,e**). A caveat with this mutation is it might disrupt an interaction with the RZ domain linker and alter its dynamics. In the paper, we also confirm CTD requirement for E2 interaction by identifying W5097A and E5108R mutants that lose E3 activity. We should also note that AlphaFold3 predicts a high confidence CTD-UBE2L3 interaction consistent with our cryoEM model.

We conclude from these findings that transthiolation has a high Michaelis constant (K_m) and activity is driven by catalytic turnover efficiency (k_{cat}). We discuss this in the revised manuscript. These data demonstrate that uncovering the catalytic nature of

C4462 and stabilizing the E2 bound complex, thereby exposing the CTD as an E2 binding site, would be extremely challenging without the ABP technology.

Supplementary Figure 4 b) Glutathione S-transferase (GST)-tagged human RNF213 CTD domain (residues 4986-5207) was expressed in *E coli*. and purified. GST-CTD pull down of UBE2L3 and a putative non-binding UBE2L3 F63A control (**Figure 4d**) was carried out. GST-CTD was bound to glutathione sepharose 4B resin and proteins were eluted with 1X LDS SDS-PAGE sample buffer. No binding was detected. Proteins were visualized by Coomassie staining. **c)** For biophysical measurements the GST tag was cleaved from GST-CTD and the protein purified. **d)** Isothermal titration calorimetry (ITC) was used to measure binding between the human CTD domain and UBE2L3. UBE2L3 was placed in the cell (70 μM) and CTD was placed in the syringe (715 μM). No binding was detected, indicating it lacks a measurable enthalpy change or the dissociation constant (K_d) is $> \sim 500 \mu\text{M}$ (based on simulated isotherms using the experimental concentrations). **e)** Pull down of the E2 UBE2D3 and a stabilized mimic of its E2~Ub conjugate by full length mouse StrepII-tagged RNF213 was carried out. Proteins were combined and RNF213 was immobilized on streptactin resin. To elute RNF213, the resin was incubated with precision protease. No pull down was detected, even in the presence of nucleotides (5 mM). Asterisks correspond to contaminants in the PreScission protease sample. **f)** Pull down of UBE2D3 or

UBE2D3_{isopeptide}~Ub following rhinovirus 3C protease treatment was also assessed by immunoblotting against UBE2D3. No pulldown was detected. The last lane is a reference consisting of a mixture of UBE2D3 and UBE2D3_{isopeptide}~Ub representing stoichiometric binding (62 and 88 ng, respectively).

Overall, our mutational analysis thus confirms the structural data of the RNF213-Ub-E2 complex, showing multivalent binding of the E2 within the E3 module.

6) The manuscript has a section titled 'RNF213 is the only ATP-regulated E3 Ub ligase' yet in the discussion (lines 390/391) the authors speculate that ZNFX1 shares a similar ATP-dependent mechanism as RNF213. Please clarify.

We have clarified in the manuscript that RNF213 is currently the only E3 ligase demonstrated to operate in an ATP-dependent manner. For ZNFX1, such regulation has only been inferred based on its domain architecture, which combines an ATP-dependent helicase domain with the RZ E3 motif. We have revised the Discussion to more explicitly state the speculative nature of the proposed ZNFX1 mechanism and to distinguish it clearly from the experimentally validated ATP-dependent activity of RNF213.

We are delighted that Reviewers 2, 3, and 4 are now fully supportive of our manuscript and recognize the significance of our findings. We greatly appreciate their positive feedback, particularly their recognition of the novel ATP regulatory mechanism, the characterization of the E2 binding module of RNF213, and the broader implications of our work.

We also thank Reviewer 1 for the thorough assessment and for bringing up further points for consideration. While this reviewer acknowledges the depth and breadth of our revisions, specific concerns are raised regarding the localization of active RNF213 and the biological source of ATP. Below, we provide detailed responses addressing each of these points.

Ahel et al have revised their manuscript entitled “ATP functions as a pathogen-associated molecular pattern to activate the E3 ubiquitin ligase RNF213.” The authors use an activity-based probe to identify ATP-dependent enzymes using mass spectrometry. They turn their focus to the large antimicrobial E3 enzyme RNF213 and do a number of assays that show that its auto-ubiquitination and LPS modifying ability requires ATP. They solve a cryo-EM structure of the complex that reveals new information about E2 binding and activity.

The reviewers asked for a number of experiments concerning the structural effects of ATP binding, the localization of active RNF213, whether ISGylation controls oligomerization and whether activity correlates with docking on bacteria or lipid droplets.

The authors addressed the following experiments:

- Creating a Cys4462Ser (C4462S) mutant and evaluated its auto-ubiquitination and lipid transfer activity.
- Knocking out Ube1L and measured ATP levels in these cells and found that ATP production was lower in Ube1L-depleted cells compared to controls
- Showed ISGylation seemed to be critical for RNF213 ATP activity

We would like to mention that the following experiments were also performed:

- Demonstrated that ATP hydrolysis by recombinant RNF213 counteracts ATP-dependent activation.
- Solved 2 new cryo-EM structures of RNF213 in order to understand how ATP binding activates the E3, which revealed 1) local conformational changes in the AAA region upon ATP binding and 2) the distinct nucleotide binding preferences of the 2 functional ATPase domains (AAA3 and AAA4).
- Used size-exclusion chromatography to look for evidence for ATP-induced higher order assembly *in vitro* but found no supporting evidence for this. We do not discount a possibility that this occurs in cells but rather is not an inherent property of the protein we have been using in our biochemical assays.
- Provided evidence that the isolated RING domain is not an active E3 *in vitro*, supporting previous data from us and others.
- Used various approaches (isothermal titration calorimetry, affinity pull-downs) to measure binding of E2 or E2~Ub to isolated RNF213 CTD, but found the interaction was too weak to measure.
- Attempted to observe endogenous RNF213 recruitment to lipid droplets by immunofluorescence microscopy in macrophages but were unable to do so.

Overall, the authors did a lot of experiments adding to the depth and breadth of the manuscript but the experiments on the localization of active RNF213 beg to be attempted with the tools the authors have developed. They say that the ABP does not provide localization but based on its biotin signal could the authors not use a fluorescently tagged streptavidin to localize active RNF213?

We thank the reviewer for this suggestion. However, determining the subcellular localization of active RNF213 using the ABP is technically challenging due to the broad reactivity of the

probe with numerous E3 ligases, as demonstrated in Fig. 3k and in a previous proteomics study (*PubMed ID: 29643511*), which identified dozens of E3 ligases labelled by the ABP. Moreover, the time frame of our experiment is very short – we determine RNF213 E3 activity 1 h after ABP electroporation. In this time frame, the electroporated cells are unable to adhere to the coverslip for a fluorescent microscopy assay. Longer incubations (12 h) for cells to properly adhere, and the ABP has been degraded. Collectively, these technical difficulties confound the ability of biotin localisation to specifically inform on the localisation of active RNF213 in our assays. An experiment where a specific signal is generated for ABP-labelled RNF213 would be necessary. We have historically explored PLA for this purpose, as suggested for a different experiment below, but this proved unsuccessful. We are in fact working on technologies that will address this but this is an independent project.

What about PLA as an alternate approach (for RNF213 and ISG15)?

We thank the reviewer for the suggestion. It is not clear to us how visualising RNF213 and ISG15 at the same site in the cell would tell us that RNF213 is modified with ISG15. Moreover, a positive PLA signal could not tell us whether the E3 is active or inactive. It is for these reasons that we have not attempted to define the site of RNF213 activity using the ABP, but rather, whether RNF213 is regulated by ATP on a cellular level, without relying on substrate modification, subcellular relocalisation, or substrate abundance as a proxy readout for E3 activity.

They do not show their immunofluorescence staining to the reviewer as well, which would have been helpful, though in the manuscript the experiments are discussed but data is not shown.

We do apologize for not including our immunofluorescence data in the initial response, we perhaps erroneously felt that it did not add anything useful to the conversation. We include

this now below, which in one experiment indeed suggests that RNF213 is activated upon recruitment to cytosolic target (bacterial) membrane. Our primary goal in this study has been – and still is – to demonstrate the regulation of RNF213 by ATP in a cellular context. While we fully acknowledge the interest of this reviewer in more detailed localization data, our focus has been on establishing ATP as a pathogen-associated molecular pattern (PAMP) and characterizing its role in RNF213 activation. Investigating the precise subcellular localization of ATP-dependent RNF213 activity, or its interplay with specific innate immune responses such as ISG15 signalling, extend beyond the scope of this study and represent important avenues for future research. This point was already addressed in our previous rebuttal letter, where we responded to the original request as follows:

Where is the activated RNF213 localized? Does changing its activity relocate it from lipid droplets to cytosol or change its oligomerization state?

“We don’t yet have a good understanding of where activated RNF213 is localised, and we have been unable to visualise recruitment of endogenous RNF213 to lipid droplets in these differentiated THP-1 cells. Please note that in our ABP assays, we only obtain information on the bulk activation status of RNF213 in a cell lysate (containing both activated and latent RNF213), while spatial information is lost. This would require specific fractionation approaches and/or design of ABPs that demonstrate fluorescence upon RNF213 labelling, which is a challenging project by itself, beyond the scope of this work. The same holds true for connecting activity status and oligomerisation behaviour in cells, which is complicated by the fact that IFN itself elevates ATP levels and thus activation of RNF213, while in parallel inducing formation of higher order RNF213 forms. Thus, it is extremely challenging to distinguish between correlation and causality.”

As our cellular activity profiling was originally performed in THP-1 cells and we were asked to visualise where activated RNF213 localises, or whether it changes its activity state from cytosol to target, or oligomerisation state, we have performed the following experiments.

- 1) We treated differentiated THP-1 cells, with or without treatment with oleic acid, or IFN alpha, for 16 h, and monitored endogenous RNF213 (as we do in our cellular ABP assays, by Western blot or mass spectrometry) using the HPA026790 rabbit polyclonal antibody, lipid droplets with the neutral lipid marker BODIPY 558/568 C12, DNA with Hoechst, by immunofluorescence microscopy. **Lipid droplets are in green, RNF213 in red, DNA in blue.** However, under conditions where we know we have activated the total pool of RNF213 (manuscript Fig. 3h), we see no colocalization of endogenous RNF213 to lipid droplets.

THP-1
RNF213 (HPA026790) - Red
BODIPY- - Green
DNA - Blue

- 2) To confirm antibody specificity for endogenous RNF213, we used a now well-established assay, infecting A549 cells with *Salmonella* Typhimurium (*S. Tm*) for 3.5 h, fixed the cells and stained for RNF213 using the same antibody HPA026790, and ubiquitin (FK2 clone). **RNF213 is in green, ubiquitin in red, DNA in blue.** Here, we do see recruitment of endogenous RNF213 to the surface of bacteria. The concomitant ubiquitin coat suggests to us that, indeed, RNF213 has become activated upon target engagement, as proposed by the reviewer. However, for the aforementioned reasons, we cannot use our ABP tool for this assay, so we prefer to interpret substrate ubiquitination as a proxy for RNF213 E3 activity with caution.

- 3) To extrapolate this assay into our THP-1 macrophages, we infected differentiated THP-1 cells with *S. Tm* for 3.5 h. **RNF213 is in green, bacteria in yellow, LAMP1 in purple, DNA in blue.** While we achieve robust infection, we are unable to visualise RNF213 recruitment to bacteria. We imagine that in these professional phagocytes, the bacteria reside in *Salmonella*-containing vacuoles (SCVs) and do not enter the cytoplasm, evidenced by colocalization with LAMP1-positive structures (e.g. vacuoles, SCVs).

THP-1 + *S. Tm*
S. Tm - Yellow
RNF213 (HPA026790) - Green
LAMP1 - Purple
DNA - Blue

Furthermore, ISGylated RNF213 was detected in THP-1 cells by They and colleagues only after enriching lipid droplets. Could the authors enrich for lipid droplets in order to visualize ISGylated RNF213 following the method that They et al used?

We appreciate the reviewer's suggestion. However, it is unclear to us why we should repeat the experiment performed by They et al., who already demonstrated that ISGylated RNF213 can be detected in lipid droplet fractions. Given that our study focuses on ATP as a PAMP and its role in RNF213 activation, further exploration of ISGylation in lipid droplets and its impact on the structure and activity of the E3 ligase is beyond the scope of our work.

It seems like the authors hypothesize that local ATP concentrations generated by bacteria could activate RNF213 but right now the underlying data to support their findings are lacking.

We acknowledge that we did not test this hypothesis by further experiments, as it was introduced purely in the discussion. A key purpose of the discussion section is to propose testable hypotheses that can guide future research, rather than only reiterate results presented in the study. Specifically, we stated:

“The finding that RNF213 E3 activity is tuned to sense cellular ATP levels during an interferon response provides important clues to its broad antimicrobial sensitivity (13,21-24). Moreover, certain RNF213-sensitive bacteria, including Salmonella, which secretes ATP into its surrounding milieu (73,74), and Chlamydia, which causes a spike in the host cell ATP/ADP ratio (75), could activate RNF213 in proximity to intracellular, metabolically active bacterial cells.”

The data concerning ISGylation could instead support the hypothesis of ISG15 regulated complex formation so by not being able to localize activity it is hard to extrapolate the biological implications of the finding.

We agree entirely with the reviewer. Our data, that ISG15 is important for RNF213 ATP-dependent regulation, could reflect either a structural requirement for ISG15 or a mechanistic role in RNF213 function (for example, ISG15's involvement in ATP metabolism) – two exciting, and not mutually exclusive, scenarios that are both supported by published literature. However, ISG15 is not the central focus of our study. Investigating its precise role in RNF213 activity, which is subject to transient changes and complex interactions with the ubiquitin-proteasome system, as well as regulation by other post-translational modifications such as phosphorylation, represents a distinct research direction. Addressing these questions would require dedicated tools tailored specifically for this purpose, making it a project of its own beyond the scope of our current study.

Since the structures unfortunately did not trap bound ATP in part of RNF213 that was fully resolvable, nor did ATP binding provoke a large conformational shift in the structure it is hard to reconcile with the primary message of the paper.

We acknowledge that this is an extremely complicated system to work with, and despite our best efforts, from some of the most experienced structural biologists in the field, we have as of yet been unable to provide a complete structural understanding of how nucleotide regulates RNF213 E3 activity. As discussed in the paper, the activation mechanism is complex, likely relying on different bound nucleotides to AAA3 and AAA4 as has been also reported for the functional switch in dynein. We would like to kindly point out that the dynein field took >10 years to uncover deeper mechanistic understanding, and even now the sequence of events coupling ATP binding/hydrolysis in dynein molecular dynamics remains unresolved. Furthermore, the how ATP binding to its AAA3 and AAA4 sites remains poorly understood. A further challenge in both cases is that the activated state is only transiently formed and thus difficult to capture and analyse by structural methods.

From an outside perspective it seems like the strength of the paper lies in the new technique to characterize E3 enzymes and less on the structural role that ATP binding plays.

With all respect, we disagree with this conclusion, as it does not fully reflect the key findings of our study. While we appreciate the positive remarks on our methodology, we believe that the mechanistic role of ATP binding in regulating E3 function, related to a unique active site and ubiquitin ligase family, is a central aspect of our work. Several key findings that support this have been omitted in this statement. To clarify, we would like to reiterate a summary from our original cover letter:

“Our work represents the first mechanistic description of the largest E3 ubiquitin ligase in the genome. Rather than focus on the antimicrobial function and lipid-specificity of RNF213 (Otten, Randow and colleagues, Nature, 2021), we asked: why have a protein that possess both AAA ATPase and E3 Ub ligase domains (not seen anywhere else in the genome)? We find that RNF213 represents the first known ATP-activated E3. Importantly, ATP regulates the upstream E3 activity and would therefore control the ubiquitination of all RNF213 substrates (lipids or proteins). Moreover, we solve a >600 kDa structure of RNF213 bound to its E2 enzyme cofactor; we map the cryptic active site cysteine using chemical biology and mass spectrometry; we develop a new, highly translatable methodology for the profiling of E3 ligase activity status inside living cells (which is already of great interest to the PROTAC and targeted protein degradation community), and we also show that interferons ubiquitously cause cellular ATP levels to rise inside macrophages, which post-translationally regulates RNF213 E3 activity inside cells. This was a truly cross-disciplinary effort from three labs (Dundee, Vienna, Glasgow) of which we are immensely proud; we believe that our data will appeal to several interrelated fields.

Our work also proposes a bigger picture view of the interferon system, which so often focuses on interferon-sensitive genes as being the root cause of the antimicrobial effects of IFN. The finding that ATP is regulated by interferon stimulation suggests to us that RNF213 (and other

enzymes?) could be regulated by pathogens, but in a pathogen-independent manner. Such a mechanism has not been described before, and we note that other groups have previously reported that the antiviral effects of IFN can be diminished by inhibiting ATP production.”